# Dissecting cell-type-specific metabolism in pancreatic ductal adenocarcinoma

**Allison N Lau[1], Zhaoqi Li[1], Laura V Danai[1,2], Anna M Westermark[1], Alicia M Darnell[1], Raphael Ferreira[1,3], Vasilena Gocheva[1], Sharanya Sivanand[1], Evan C Lien[1], Kiera M Sapp[1], Jared R Mayers[1], Giulia Biffi[4,5,6], Christopher R Chin[1], Shawn M Davidson[1,7,8], David A Tuveson[4,5], Tyler Jacks[1], Nicholas J Matheson[1,9,10], Omer Yilmaz[1,11], Matthew G Vander Heiden[1,12]\***

[1]Koch Institute for Integrative Cancer Research and the Department of Biology at Massachusetts Institute of Technology, Cambridge, United States; [2]Department of Biochemistry and Molecular Biology, University of Massachusetts, Amherst, Amherst, United States; [3]Department of Biology and Biological Engineering, Chalmers University of Technology, Gothenburg, Sweden; [4]Cold Spring Harbor Laboratory, Cold Spring Harbor, New York, United States; [5]Lustgarten Foundation Pancreatic Cancer Research Laboratory, Cold Spring Harbor, New York, United States; [6]Cancer Research United Kingdom Cambridge Institute, University of Cambridge, Cambridge, United Kingdom; [7]Lewis-Sigler Institute for Integrative Genomics, Princeton University, Princeton, United States; [8]Department of Molecular Biology, Princeton University, Princeton, United States; [9]Department of Medicine, University of Cambridge, Cambridge, United Kingdom; [10]Cambridge Institute for Therapeutic Immunology and Infectious Disease, University of Cambridge, Cambridge, United Kingdom; [11]Department of Pathology, Massachusetts General Hospital, Boston, United States; [12]Department of Medical Oncology, Dana-Farber Cancer Institute, Boston, United States

*For correspondence:
mvh@mit.edu

**Abstract** Tumors are composed of many different cell types including cancer cells, fibroblasts, and immune cells. Dissecting functional metabolic differences between cell types within a mixed population can be challenging due to the rapid turnover of metabolites relative to the time needed to isolate cells. To overcome this challenge, we traced isotope-labeled nutrients into macromolecules that turn over more slowly than metabolites. This approach was used to assess differences between cancer cell and fibroblast metabolism in murine pancreatic cancer organoid-fibroblast co-cultures and tumors. Pancreatic cancer cells exhibited increased pyruvate carboxylation relative to fibroblasts, and this flux depended on both pyruvate carboxylase and malic enzyme 1 activity. Consequently, expression of both enzymes in cancer cells was necessary for organoid and tumor growth, demonstrating that dissecting the metabolism of specific cell populations within heterogeneous systems can identify dependencies that may not be evident from studying isolated cells in culture or bulk tissue.

## Introduction

Tumors are composed of a heterogeneous mix of cell types, including cancer cells and stromal cells such as fibroblasts, macrophages, and other immune cells. How these different cell types interact to enable tumor growth is poorly understood. Environmental context plays an important role in determining how cancer cells use nutrients to proliferate and survive, and non-cancer cells within a tissue can alter nutrient availability (*Lyssiotis and Kimmelman, 2017*; *Mayers and Vander Heiden, 2017*;

**eLife digest** Tumors contain a mixture of many different types of cells, including cancer cells and non-cancer cells. The interactions between these two groups of cells affect how the cancer cells use nutrients, which, in turn, affects how fast these cells grow and divide. Furthermore, different cell types may use nutrients in diverse ways to make other molecules – known as metabolites – that the cell needs to survive.

Fibroblasts are a subset of non-cancer cells that are typically found in tumors and can help them form. Separating fibroblasts from cancer cells in a tumor takes a lot longer than the chemical reactions in each cell of the tumor that produce and use up nutrients, also known as the cell's metabolism. Therefore, measuring the levels of glucose (the sugar that is the main energy source for cells) and other metabolites in each tumor cell after separating them does not necessarily provide accurate information about the tumor cell's metabolism. This makes it difficult to study how cancer cells and fibroblasts use nutrients differently.

Lau et al. have developed a strategy to study the metabolism of cancer cells and fibroblasts in tumors. Mice with tumors in their pancreas were provided glucose that had been labelled using biochemical techniques. As expected, when the cell processed the glucose, the label was transferred into metabolites that got used up very quickly. But the label also became incorporated into larger, more stable molecules, such as proteins. Unlike the small metabolites, these larger molecules do not change in the time it takes to separate the cancer cells from the fibroblasts.

Lau et al. sorted cells from whole pancreatic tumors and analyzed large, stable molecules that can incorporate the label from glucose in cancer cells and fibroblasts. The experiments showed that, in cancer cells, these molecules were more likely to have labeling patterns that are characteristic of two specific enzymes called pyruvate carboxylase and malic enzyme 1. This suggests that these enzymes are more active in cancer cells. Lau et al. also found that pancreatic cancer cells needed these two enzymes to metabolize glucose and to grow into large tumors.

Pancreatic cancer is one of the most lethal cancers and current therapies offer limited benefit to many patients. Therefore, it is important to develop new drugs to treat this disease. Understanding how cancer cells and non-cancer cells in pancreatic tumors use nutrients differently is important for developing drugs that only target cancer cells.

*Muir et al., 2018*; *Pavlova and Thompson, 2016*; *Sullivan and Vander Heiden, 2019*). There is evidence that different cell populations within tumors can compete for limiting nutrients (*Chang et al., 2015*; *Ho et al., 2015*; *Zecchin et al., 2017*), and metabolic cooperation between different cell types can also influence tumor phenotypes (*Linares et al., 2017*; *Sousa et al., 2016*; *Valencia et al., 2014*; *Vander Heiden and DeBerardinis, 2017*). Nevertheless, technical challenges associated with studying the metabolism of individual cell types within a mixed population have limited a complete understanding of the metabolic interactions between cells in tumors. More broadly, this challenge has been a barrier to study how cells use nutrients differently within tissues to support both normal and disease physiology.

Cancer cell metabolism in culture can differ from the metabolism of tumors in vivo (*Biancur et al., 2017*; *Cantor et al., 2017*; *Davidson et al., 2016*; *Mayers and Vander Heiden, 2015*; *Muir et al., 2017*; *Sellers et al., 2019*; *Vande Voorde et al., 2019*). This can at least partially be ascribed to changes in cancer cell metabolism that are driven by different nutrients present in the extracellular environment; however, another major difference between cell culture and tumors is the presence of additional cell types within tumors that are absent from most culture systems. The presence of many different cell types complicates the ability to characterize cancer cell metabolism in tumors, particularly in cases where a minority of the tumor is composed of cancer cells. For instance, cancer cells are a minority cell type in pancreatic ductal adenocarcinoma (PDAC) tumors (*Feig et al., 2012*), and an understanding of cell metabolism in these tumors requires de-convolution of cancer-specific and stroma-specific phenotypes. Furthermore, there is evidence that the metabolism of cancer cells and different stromal cells isolated from these tumors can be different from each other when studied in culture (*Francescone et al., 2018*; *Halbrook et al., 2019*; *Sousa et al., 2016*), and it is unknown

whether the metabolic programs used by different cell populations in culture are also used within PDAC tumors in vivo where environmental conditions are different.

Studies of bulk tumor metabolism fail to capture information about metabolic heterogeneity with regard to different cell types (*Xiao et al., 2019*), and existing approaches are limited in their ability to study functional metabolic phenotypes in different cell populations in intact tissue. A major limitation arises from the fact that metabolic reactions take place on time scales that are faster than the turnover of many metabolic intermediates, complicating metabolite analysis after tumor digestion and cell sorting (*Shamir et al., 2016*). Furthermore, cell sorting exposes cells to conditions that are different from those experienced by cells in tissues and can change metabolism in many ways. For example, sorting can induce mechanical and oxidative stress and reduce the levels of certain metabolites (*Binek et al., 2019*; *Llufrio et al., 2018*; *Roci et al., 2016*), and even adding small amounts of bovine serum to the sorting buffer was not sufficient to prevent changes in some metabolite levels during cell sorting (*Llufrio et al., 2018*). Indeed, isotopic metabolite labeling patterns can be more robust than metabolite levels when assessing metabolites from flow cytometry sorted cells, although labeling can also be affected by cell sorting (*Roci et al., 2016*). Nevertheless, interpretation of metabolite labeling patterns is influenced by whether cells are at metabolic steady state (*Buescher et al., 2015*), which when coupled with the rapid timescales of metabolism relative to the time needed to isolate cells suggests that new approaches are needed to better understand metabolism of individual cell types within mixed cell populations such as tumors.

To overcome the challenges associated with studying cell metabolism within intact tumors and organoid co-cultures, we adapted an approach based on end-product biomass labeling (*Green et al., 2016*; *Hosios et al., 2016*; *Le et al., 2017*; *Lewis et al., 2014*; *Mayers et al., 2016*; *Shankaran et al., 2016*). This technique has been applied in studies of microbial metabolism and metabolic engineering to better understand mixed populations of bacteria (*Gebreselassie and Antoniewicz, 2015*; *Ghosh et al., 2014*; *Rühl et al., 2011*; *Zamboni et al., 2005*). Unlike short-lived metabolic intermediates, turnover of end-product macromolecules such as protein is slow relative to the time period needed to isolate tumor cell populations. By assessing the isotope-labeling pattern of end-product biomass generated when cells are exposed to labeled nutrients in a mixed cell population, metabolic differences in nutrient use by different cells can be inferred. We used this approach to uncover a difference in glucose metabolism between cancer cells and fibroblasts in PDAC. Specifically, we find that, relative to fibroblasts, cancer cells within PDAC tumors have increased use of glucose for tricarboxylic acid (TCA) cycle anaplerosis through increased flux through pyruvate carboxylase (PC). This phenotype is not evident when cancer cells and fibroblasts are studied as separate populations in mono-culture, even though PC is necessary for tumor growth in vivo. Furthermore, deletion of PC was insufficient to account for all pyruvate carboxylation activity within cancer cells in a mixed population, revealing that malic enzyme 1 (ME1) also contributes to pyruvate carboxylation in cancer cells when fibroblasts are present and is required for PDAC tumor growth. These data argue that tracing labeled nutrients into stable biomass can be used to reveal metabolic differences between subpopulations of cells in a mixed cell system and to identify phenotypes that depend on the co-existence of multiple cell types.

## Results

### Glucose metabolism in pancreatic tumors

PDAC involves tumors where cancer cells can be a minority cell population (*Feig et al., 2012*). To better understand glucose metabolism of PDAC tumor tissue in vivo, we infused U-$^{13}$C-glucose into conscious, unrestrained mice (*Davidson et al., 2016*; *Hui et al., 2017*; *Marin-Valencia et al., 2012*) bearing PDAC tumors from autochthonous models that are driven by activating mutations in *Kras* and disruption of *Trp53* function (*Bardeesy et al., 2006*; *Hingorani et al., 2005*). Similar to what has been observed with other mouse cancer models and in humans (*Davidson et al., 2016*; *Fan et al., 2009*; *Hensley et al., 2016*; *Sellers et al., 2015*), extensive labeling of multiple metabolic intermediates is observed from U-$^{13}$C-glucose in pancreatic tumors and normal pancreas (*Figure 1*, *Figure 1—figure supplements 1–3*, *Figure 1—source data 1*, *Figure 1—figure supplement 1—source data 1*, *Figure 1—figure supplement 2—source data 1*, *Figure 1—figure supplement 3—source data 1*).

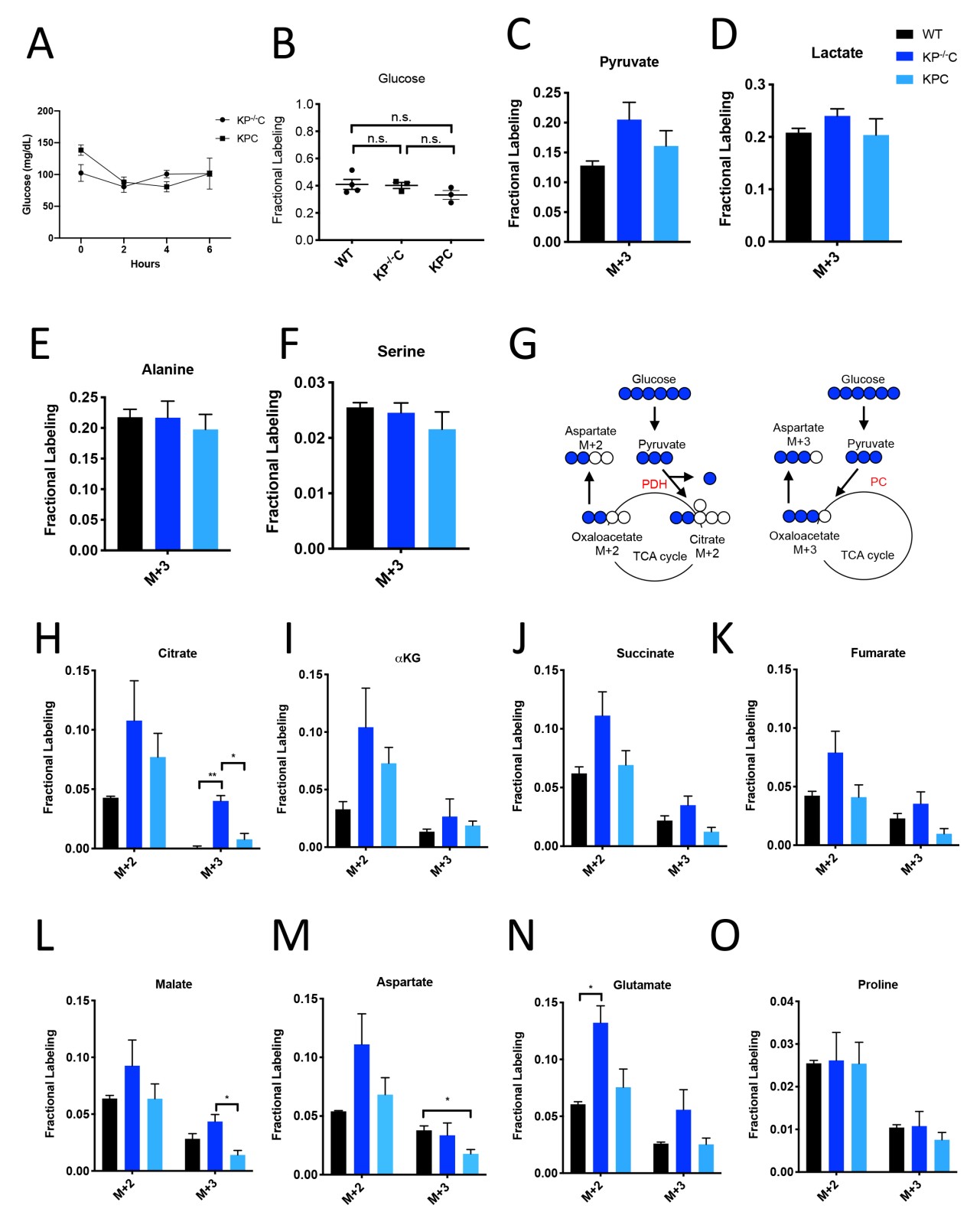

**Figure 1.** Glucose metabolism in PDAC tumors. (**A**) Plasma glucose levels over time in autochthonous *LSL-Kras^G12D/+*; *Trp53^fl/fl*; *Pdx1-Cre* (KP^-/-^C) or autochthonous *LSL-Kras^G12D/+*; *Trp53^R172H/+*; *Pdx1-Cre* (KPC) pancreatic tumor-bearing mice infused with U-$^{13}$C-glucose at a rate of 0.4 mg/min. n = 3 for each group. Mean +/- SEM is shown. (**B**) Enrichment of fully labeled glucose (M+6) in plasma from the indicated mice following a 6 hr U-$^{13}$C-glucose infusion at a rate of 0.4 mg/min. Non-tumor bearing C57Bl6/J (WT) mice were used to assess metabolite labeling in normal pancreas. WT, n = 4; KP^-/-^C,
*Figure 1 continued on next page*

Figure 1 continued

n = 3, KPC, n = 3. Differences in plasma glucose enrichment were not significant between WT and KP$^{-/-}$C mice (p=0.8723), WT and KPC mice (p=0.1907), or KP$^{-/-}$C and KPC tumor-bearing mice (p=0.1512) based on unpaired, two-tailed student's t tests. Mean +/- SEM is shown. (C–F) The fractional labeling of pyruvate (C), lactate (D), alanine (E), and serine (F) in pancreas (black), autochthonous KP$^{-/-}$C pancreatic tumors (dark blue), or autochthonous KPC pancreatic tumors (light blue) following a 6 hr U-$^{13}$C-glucose infusion at a rate of 0.4 mg/min. The M+3 isotopomers are shown for each metabolite: n = 3 for each group. Mean +/- SEM is shown. (G) Schematic illustrating how U-$^{13}$C-glucose can label TCA cycle intermediates. An M+2 labeling pattern of TCA cycle intermediates can be derived from flux through pyruvate dehydrogenase (PDH) (left) while an M+3 labeling pattern can reflect flux through pyruvate carboxylase (PC) (right). (H–O) The fractional labeling of citrate (M+3; WT vs. KP$^{-/-}$C p=0.0012, KP$^{-/-}$C vs. KPC p=0.0084) (H), α-ketoglutarate (αKG) (I), succinate (J), fumarate (K), malate (M+3 KP$^{-/-}$C vs. KPC p=0.0156) (L), aspartate (M+3 WT vs. KPC p=0.0194) (M), glutamate (M+2 WT vs. KP$^{-/-}$C p=0.0089) (N), and proline (O) in pancreas (black), autochthonous KP$^{-/-}$C pancreatic tumors (dark blue), or autochthonous KPC pancreatic tumors (light blue) following a 6 hr U-$^{13}$C-glucose infusion at a rate of 0.4 mg/min. Significance based on unpaired, Students t-test. The M+2 and M+3 isotopomers are shown for each metabolite, n = 3 for each group. Mean +/- SEM is shown.

The online version of this article includes the following source data and figure supplement(s) for figure 1:

**Source data 1.** Isotope labeling of tumors in U-$^{13}$C- glucose-infused mice with autochthonous PDAC tumors presented in *Figure 1*.

**Figure supplement 1.** Metabolite abundance and plasma labeling in U-$^{13}$C- glucose-infused autochthonous PDAC tumors.

**Figure supplement 1—source data 1.** Tumor metabolite abundance and plasma isotope labeling in U-$^{13}$C- glucose-infused mice with autochthonous PDAC tumors presented in *Figure 1—figure supplement 1*.

**Figure supplement 2.** Glucose metabolism in autochthonous PDAC tumors infused with U-$^{13}$C- glucose at 30 mg/kg/min.

**Figure supplement 2—source data 1.** Isotope labeling of tumors in U-$^{13}$C- glucose-infused mice with autochthonous PDAC tumors presented in *Figure 1—figure supplement 2*.

**Figure supplement 3.** Glucose metabolism in autochthonous and orthotopic PDAC tumors infused with U-$^{13}$C-glucose at 30 mg/kg/min.

**Figure supplement 3—source data 1.** Plasma isotope labeling in $^{13}$C- glucose-infused mice with autochthonous PDAC tumors and metabolite isotope labeling in orthotopic PDAC tumors presented in *Figure 1—figure supplement 3*.

To assess U-$^{13}$C-glucose labeling of metabolites in autochthonous pancreatic tumors arising in the *LSL-Kras$^{G12D}$*; *Trp53$^{fl/fl}$*; *Pdx1-Cre* (KP$^{-/-}$C) (*Bardeesy et al., 2006*) and the *LSL-Kras$^{G12D}$*; *Trp53$^{R172H/+}$*; *Pdx1-Cre* (KPC) (*Hingorani et al., 2005*) mouse models, we first confirmed that plasma glucose levels were not changed over the course of the experiment (*Figure 1A*). At this rate of glucose infusion, enrichment of $^{13}$C-glucose in plasma was around 40% in non-tumor-bearing mice, and in both KP$^{-/-}$C and KPC animals (*Figure 1B*). Glucose enrichment was not measured in tumors, as glucose is thought to be rapidly metabolized after entry into cells (*Nguyen et al., 2011*; *Parikh et al., 2015*; *Yeh et al., 2018*). Under these conditions, labeling of pyruvate and lactate, as well as the glucose-derived amino acids alanine and serine, from U-$^{13}$C-glucose was observed in normal pancreas tissue and in tumors arising in both models (*Figure 1C–F*, *Figure 1—source data 1*).

Labeling of TCA cycle metabolites was also observed in normal pancreas tissue and in PDAC tumors from both models (*Figure 1G–L*, *Figure 1—source data 1*). Carbon from glucose-derived pyruvate can contribute to the TCA cycle via reactions catalyzed by pyruvate dehydrogenase (PDH) or pyruvate carboxylase (PC), and the relative use of these routes of TCA cycle labeling has been inferred from the labeling pattern of TCA cycle intermediates from $^{13}$C-labeled glucose (*Davidson et al., 2016*; *Fan et al., 2009*; *Hensley et al., 2016*; *Sellers et al., 2015*). PDH decarboxylates three-carbon pyruvate to two-carbon acetyl-CoA, and therefore if $^{13}$C-labeled pyruvate is metabolized via PDH, a two carbon (M+2) labeling pattern is observed in TCA cycle metabolites as well as in the amino acid aspartate (*Figure 1G*). In contrast, PC carboxylates three-carbon pyruvate to four-carbon oxaloacetate, and therefore if unlabeled $CO_2$ is added to $^{13}$C-labeled pyruvate via this enzyme, a three carbon (M+3) labeling pattern is observed (*Figure 1G*). We observed an increase in both M+2 and M+3 labeling of TCA cycle metabolites and the TCA cycle-derived amino acids aspartate and glutamate in KP$^{-/-}$C tumor tissue compared to normal pancreas (*Figure 1H–N*). Importantly, despite some aspartate labeling in the plasma of infused mice (*Figure 1—figure supplement 1J*, *Figure 1—figure supplement 1—source data 1*), prior studies of this model have shown that cancer cells cannot take up extracellular aspartate (*Sullivan et al., 2018*), arguing that labeled aspartate in plasma contributes minimally to aspartate labeling in cancer cells in PDAC tumors. Proline was not extensively labeled from glucose in pancreas or PDAC tumor tissue from either model, suggesting that glucose carbon contributes minimally to the synthesis of this amino acid (*Figure 1O*, *Figure 1—source data 1*). Of note, in KPC mouse PDAC tumors, we observed higher M+2 labeling of only some TCA cycle metabolites compared to normal pancreas, and did not observe an increase in M+3 metabolite labeling compared to pancreas (*Figure 1H–O*), illustrating

that glucose labels metabolites differently in bulk tumors arising in each model. These data suggest that glucose contributes to labeling of TCA cycle carbon via reactions that involve PDH and PC in PDAC tumors, although the extent of labeling depends on the PDAC model used.

To further study glucose metabolism in PDAC tumor tissue, we infused control and tumor-bearing KP$^{-/-}$C mice with U-$^{13}$C-glucose at a higher rate in an attempt to increase plasma enrichment of labeled glucose (*Davidson et al., 2016*; *Figure 1—figure supplement 2*, *Figure 1—figure supplement 3A–K*, *Figure 1—figure supplement 2—source data 1*, *Figure 1—figure supplement 3—source data 1*). Plasma glucose levels were increased as a result of this higher infusion rate (*Figure 1—figure supplement 2B*), and resulted in extensive labeling of glycolytic metabolites and glucose-derived amino acids (*Figure 1—figure supplement 2C–F*, *Figure 1—figure supplement 2—source data 1*), as well as an increase in M+2 and M+3 labeling of TCA cycle intermediates and related amino acids in pancreatic tumor tissue relative to normal pancreas (*Figure 1—figure supplement 2G–N*, *Figure 1—figure supplement 2—source data 1*). These data further support that glucose carbon can contribute label to TCA cycle intermediates via pathways that involve PDH and PC in PDAC tumors, but the relative contribution varies based on plasma glucose levels and the PDAC model examined.

Despite both being driven by mutant *Kras* and loss of normal *Trp53* function, differences in the autochthonous KP$^{-/-}$C and KPC PDAC models are known and have been attributed to differences in tumor latency and p53 status, as well as differences in how stromal cell populations interact with cancer cells to support tumor growth (*Rosenfeldt et al., 2013*; *Vennin et al., 2019*). Thus, a difference in relative abundance of cancer and non-cancer cells, or in interactions between non-cancer cells and cancer cells, are possible explanations for why differences in glucose labeling are observed across these models. To explore whether the relative abundance of cancer cells in the tumor might affect labeling, we infused mice with pancreatic tumors derived from orthotopic injection of a syngeneic *Kras*$^{G12D}$; *Trp53*$^{-/-}$ pancreatic cancer cell line derived from tumors arising in the KP$^{-/-}$C model (*Danai et al., 2018*), since cell line transplantation models are thought to result in tumors with a less dense, desmoplastic stroma compared to autochthonous models (*Baker et al., 2016*; *Olive et al., 2009*). When compared to adjacent normal pancreas, we observed an increase in M+2 and M+3 labeling of TCA metabolites and aspartate in this orthotopic tumor model (*Figure 1—figure supplement 3L–O*, *Figure 1—figure supplement 3—source data 1*). Regardless, tumors arising from orthotopic transplantation of murine PDAC cells still contain stroma (*Danai et al., 2018*). Thus, tumors in all models considered consist of multiple cell types, and cancer cells are known to be a minority cell population in both autochthonous PDAC tumor models. In all cases, metabolite labeling will reflect a weighted average of labeling in all cell types present in the tissue sample and this heterogeneity in cell types in all tissues is a limitation to the use of labeled nutrient infusions to understand the metabolism of cancer cells, or any individual cell population, in tumors or other tissues.

## Assessment of pyruvate carboxylation activity in different PDAC tumor cell populations using existing methods

Dissecting the metabolism of individual cell types within a mixed cell population is a barrier to identifying cancer cell-specific liabilities via functional metabolic measurements in tumors such as PDAC. Therefore, we sought to better understand the contribution of cancer cells to the labeling of TCA cycle intermediates from $^{13}$C-glucose in PDAC tumors. We focused on M+3 labeling of TCA cycle intermediates from glucose, reflective of pyruvate carboxylation activity, because this is a metabolic phenotype observed in tumors that is less prominent in cancer cells in culture (*Davidson et al., 2016*). One existing approach to determine which cell type within the tumor contributes to this activity is to evaluate expression of an enzyme known to catalyze this reaction. Indeed, immunohistochemistry (IHC) analysis of tumors arising in KP$^{-/-}$C mice revealed higher PC expression in cancer cells (*Figure 2A*). Analysis of a human pancreatic tumor tissue by IHC shows that human tumors exhibit a range of PC expression levels (*Figure 2—figure supplement 1A–D*), although higher PC expression is observed in cancer cells compared to stroma (*Figure 2B*), similar to findings in human lung tumors (*Sellers et al., 2015*). However, while IHC can be useful to determine relative expression in tumor sections, it does not prove lack of expression by non-cancer cells. Indeed, qPCR analysis of mRNA isolated from sorted cell populations derived from KP$^{-/-}$C tumors trended toward higher PC expression in cancer cells relative to fibroblasts, although PC mRNA was detected in fibroblasts (*Figure 2C*). Furthermore, metabolic fluxes can be more dependent on metabolite concentrations

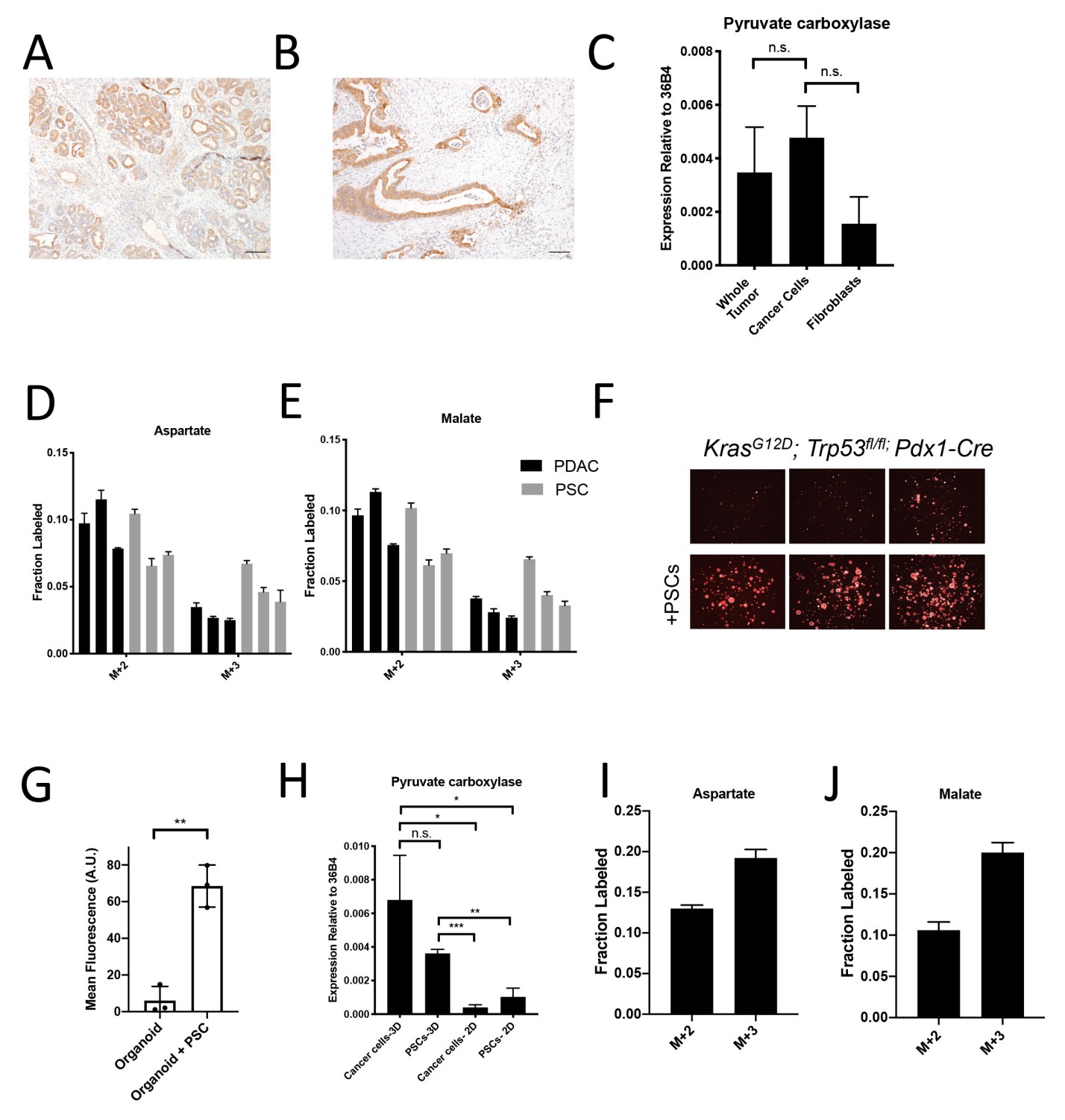

**Figure 2.** Assessment of pyruvate carboxylase activity in PDAC cancer cells and fibroblasts. (**A**) Sections from tumors arising in *LSL-Kras^G12D/+*; *Trp53^fl/fl*; *Pdx1-Cre* (KP^-/-C) mice were stained with an antibody against pyruvate carboxylase. Scale bar represents 100 μm. (**B**) Representative image from a human pancreatic tumor tissue microarray that was stained with an antibody against pyruvate carboxylase. Scale bar represents 100 μm. (**C**) Pyruvate carboxylase expression was assessed by qPCR in whole PDAC tumors and different cell populations in sorted from PDAC tumors arising KP^-/-C mice. The difference in expression between sorted cancer cells and the whole tumor (p=0.5485) or between sorted cancer cells and fibroblasts from tumors was not significant (p=0.0722) based on unpaired, two-tailed student's t-tests. Mean +/- SEM is shown. 36B4 was used as a housekeeping gene control. (**D–F**) The fractional labeling of aspartate (**D**) or malate (**E**) in cultured murine PDAC cells derived from tumors in KP^-/-C mice (black) and in cultured

*Figure 2 continued on next page*

**Figure 2 continued**

isolated pancreatic stellate cells (PSCs) (grey) was measured after exposure to U-$^{13}$C-glucose for 24 hr. The M+2 and M+3 isotopomers are shown for each metabolite. Mean +/- SD is shown. (F) Fluorescent images of KP$^{-/-}$C PDAC cancer cell organoids expressing tdTomato cultured in DMEM without pyruvate with 10% dialyzed FBS (top) or in the same media conditions with murine PSCs included in the culture (bottom). (G) Quantification of tdTomato fluorescence from images in (G). Organoid cultures with PSCs included had significantly higher tdTomato fluorescence than organoids cultured without PSCs (p=0.0014) based on an unpaired, two-tailed student's t-test. Mean +/- SD is shown. (H) Expression of pyruvate carboxylase as assessed by qPCR in PDAC cancer cells or PSC cells sorted from organoid co-cultures (3D) or in standard 2D culture as indicated. The difference between cancer cells in 3D compared to PSCs in 3D (p=0.1071) was not significant, but the differences between cancer cells in 3D and cancer cells in 2D (p=0.0140), cancer cells in 3D compared to PSCs in 2D (p=0.0209), PSCs in 3D compared to cancer cells in 2D (p<0.0001), and PSCs in 3D compared to PSCs in 2D (p=0.0014) were significant based on unpaired, two-tailed student's t-tests. Mean +/- SD is shown. (I–J) The fractional labeling of aspartate (I) or malate (J) in murine PDAC organoid-PSC co-cultures was measured after 24 hr of exposure to U-$^{13}$C-glucose. The M+2 and M+3 isotopomers are shown for each metabolite. Mean +/- SD is shown.

The online version of this article includes the following source data and figure supplement(s) for figure 2:

**Source data 1.** Metabolite isotope labeling by U-$^{13}$C- glucose in unsorted organoid-PSC co-cultures.
**Figure supplement 1.** Assessment of pyruvate carboxylase activity in PDAC cancer cells and fibroblasts.

than enzyme expression levels (*Hackett et al., 2016*). Therefore, increased enzyme expression may not reflect increased activity in tissues, and argues that while suggestive, expression analysis is not definitive for identifying which cell population is responsible for M+3 TCA cycle labeling from glucose in pancreatic tumors.

Another approach that has been used to determine which cell type(s) within the tumor contribute to a metabolic activity is isolating distinct cell populations and studying them in culture (*Dalin et al., 2019*; *Francescone et al., 2018*; *Linares et al., 2017*; *Sousa et al., 2016*; *Valencia et al., 2014*; *Yang et al., 2016*). Pancreatic stellate cells (PSCs) are a type of resident fibroblast in the pancreas which can become activated during tumorigenesis and impact the tumor microenvironment (*Bynigeri et al., 2017*; *Dunér et al., 2011*). When PDAC cells or PSCs alone are cultured in the presence of $^{13}$C-glucose in vitro, PSCs exhibit similar or higher M+3 TCA cycle metabolite labeling than cancer cells (*Figure 2D–E*), even though the fibroblast cell population exhibited lower PC expression in tumors (*Figure 2A–B*). These data further highlight the challenges associated with ascribing functional metabolic phenotypes using enzyme expression alone. Nevertheless, isolated cell populations in culture also may not retain the same functional metabolic phenotypes found within tumor tissue where many different cells compete for available nutrients.

To develop new approaches to study the phenotype of individual cell types in a mixed cell population, we first sought to generate a more tractable system that only involves interactions between two different cell types. One approach is to use organoid cultures involving PDAC cancer cells and fibroblasts (*Öhlund et al., 2017*) where nutrient conditions are modified such that cancer cells rely on the presence of the fibroblasts to proliferate. To do this, we generated pancreatic cancer organoid cultures from KP$^{-/-}$C and KPC tumors (*Figure 2—figure supplement 1E*; *Boj et al., 2015*), and found that when exposed to a more minimal medium than is commonly used (*Boj et al., 2015*), pancreatic cancer organoid growth becomes dependent on including PSCs in the culture system (*Figure 2F–G*, *Figure 2—figure supplement 1F–G*). Relevant to the M+3 labeling of TCA cycle-associated metabolites from glucose observed in pancreatic tumors, when sorted from this co-culture organoid system, both cancer cells and PSCs expressed higher levels of PC mRNA compared to PSCs and pancreatic cancer cells in standard monoculture (*Figure 2H*). In addition, when U-$^{13}$C-glucose is provided to whole organoid co-cultures comprised of both cancer cells and PSCs, and TCA cycle intermediate labeling is assessed after rapid quenching and extraction of metabolites, M+3 labeling of aspartate and malate was observed (*Figure 2I–J*, *Figure 2—figure supplement 1H–K*, *Figure 2—source data 1*). These data argue that this organoid co-culture system may provide a model to explore the relative contribution of each cell population to the pyruvate carboxylation phenotype observed when both cell types are present.

## Effect of cell sorting on metabolite levels and metabolite labeling from extracellular nutrients

To dissect PDAC cancer cell versus other cell type contributions to specific metabolic activities, we reasoned it would be necessary to isolate each cell type for analysis after exposure to labeled

nutrients. To experimentally evaluate the effect of sorting cells from the organoid co-culture system on metabolite levels and labeling from glucose, we cultured AL1376 murine PDAC cells in U-[13]C-glucose and incubated the cells in buffer on ice for various lengths of time to simulate conditions the cells would experience during separation by flow cytometry (up to 240 min) or other antibody based methods, which require a minimum of 10–12 min (*Abu-Remaileh et al., 2017*; *Chen et al., 2016*). Metabolite levels and labeling were then measured over time using mass spectrometry, allowing comparison to that observed when metabolism is rapidly quenched (the zero time point). Consistent with the known rapid turnover of metabolites (*Shamir et al., 2016*), the levels (*Figure 3A–D*, *Figure 3—figure supplement 1A–C*, *Figure 3—source data 1*) and/or labeling from U-[13]C-glucose (*Figure 3E–H*, *Figure 3—figure supplement 1D–F*, *Figure 3—source data 1*), of many metabolites changed over the time required to separate cells using antibodies and/or flow cytometry. These changes indicate that metabolism is not at metabolic steady-state where levels and labeling of metabolites are stable over time and could complicate interpretation of some differential isotope labeling patterns (*Buescher et al., 2015*). In fact, changes in metabolite levels and labeling may be even greater when using flow cytometry to sort cells in practice because temperature as well as factors such as mechanical stress are less easily controlled (*Binek et al., 2019*). While this does not absolutely preclude an ability to gain information from metabolite measurements in sorted cells, assessment of M+3 labeling of TCA cycle intermediates in sorted cell populations from organoids or tumors may not fully portray the contribution of each cell type to the pyruvate carboxylation phenotype observed when material containing multiple cell types is analyzed.

The turnover of protein and nucleic acid is slow relative to metabolites (*Shamir et al., 2016*), which allows gene expression and proteomic analysis in separated cell types to better reflect the state of cells within a mixed population. Because metabolites contribute to protein, lipid, and nucleic acid biomass, and isotope-labeled nutrients can be traced into this biomass (*Gebreselassie and Antoniewicz, 2015*; *Ghosh et al., 2014*; *Green et al., 2016*; *Hosios et al., 2016*; *Le et al., 2017*; *Lewis et al., 2014*; *Mayers et al., 2016*; *Rühl et al., 2011*; *Shankaran et al., 2016*; *Zamboni et al., 2005*), we reasoned that [13]C-labeling patterns in biomass might be used to infer the contribution of glucose to different metabolic pathways within a mixed cell population relevant to pancreatic cancer. We confirmed that glucose labeling of protein was stable over the time period needed to sort cells by flow cytometry (*Figure 3I–J*, *Figure 3—source data 1*). We also confirmed that amino acids from protein hydrolysates were detectable in sorted cells from murine PDAC tumors, and were within the linear range of detection by GC-MS even when low cell numbers of less abundant cell populations were recovered from tumors (*Figure 3—figure supplement 1G–L*). Therefore, examining [13]C label in amino acids from hydrolyzed protein may be informative of the labeling of free amino acids in tumor cell subpopulations that existed prior to sorting the cells.

## Evidence for higher pyruvate carboxylation activity in cancer cells relative to fibroblasts in PDAC models

To facilitate sorting of PSCs and PDAC cancer cells from organoid co-cultures and tumors, a *LSL-tdTomato* reporter allele was bred to the KP[-/-]C and KPC PDAC models as a source of tdTomato[+] cancer cells for both organoid and tumor models. PSCs were isolated from pancreata from mice bearing a *β-actin-GFP* allele to enable sorting of GFP[+] PSCs for labeling of the PSC population in the organoid co-culture model (*Figure 4A*). To determine the relative contribution of [13]C-glucose to M+3-labeled aspartate in cancer cells and PSCs in the organoid-fibroblast co-culture model, we exposed organoid co-cultures containing tdTomato[+] cancer cells and GFP[+] PSCs to U-[13]C-glucose for 1–4 days prior to sorting cancer cells and PSCs, and then hydrolyzed protein for amino acid analysis from each cell population. Over time, similar M+2 protein aspartate labeling was observed between the two cell types, while higher M+3 aspartate labeling was observed in cancer cells as compared to PSCs, suggesting that while the two cell types have similar labeling via reactions involving PDH, the cancer cells appear to have higher pyruvate carboxylation activity (*Figure 4B–C*, *Figure 4—source data 1*). This higher M+3 level was also reflected in the other TCA cycle-derived amino acids glutamate and proline (*Figure 4D–G*, *Figure 4—source data 1*), whereas labeling of the glucose-derived amino acids alanine and serine was not higher in cancer cells (*Figure 4—figure supplement 1A–B*, *Figure 4—source data 1*). We also exposed organoid co-cultures to U-[13]C-glutamine over 4 days and traced the fate of labeled carbon into protein in each cell population. Of note, we observed slightly higher labeling of aspartate from glutamine in protein in PSCs (*Figure 4—*

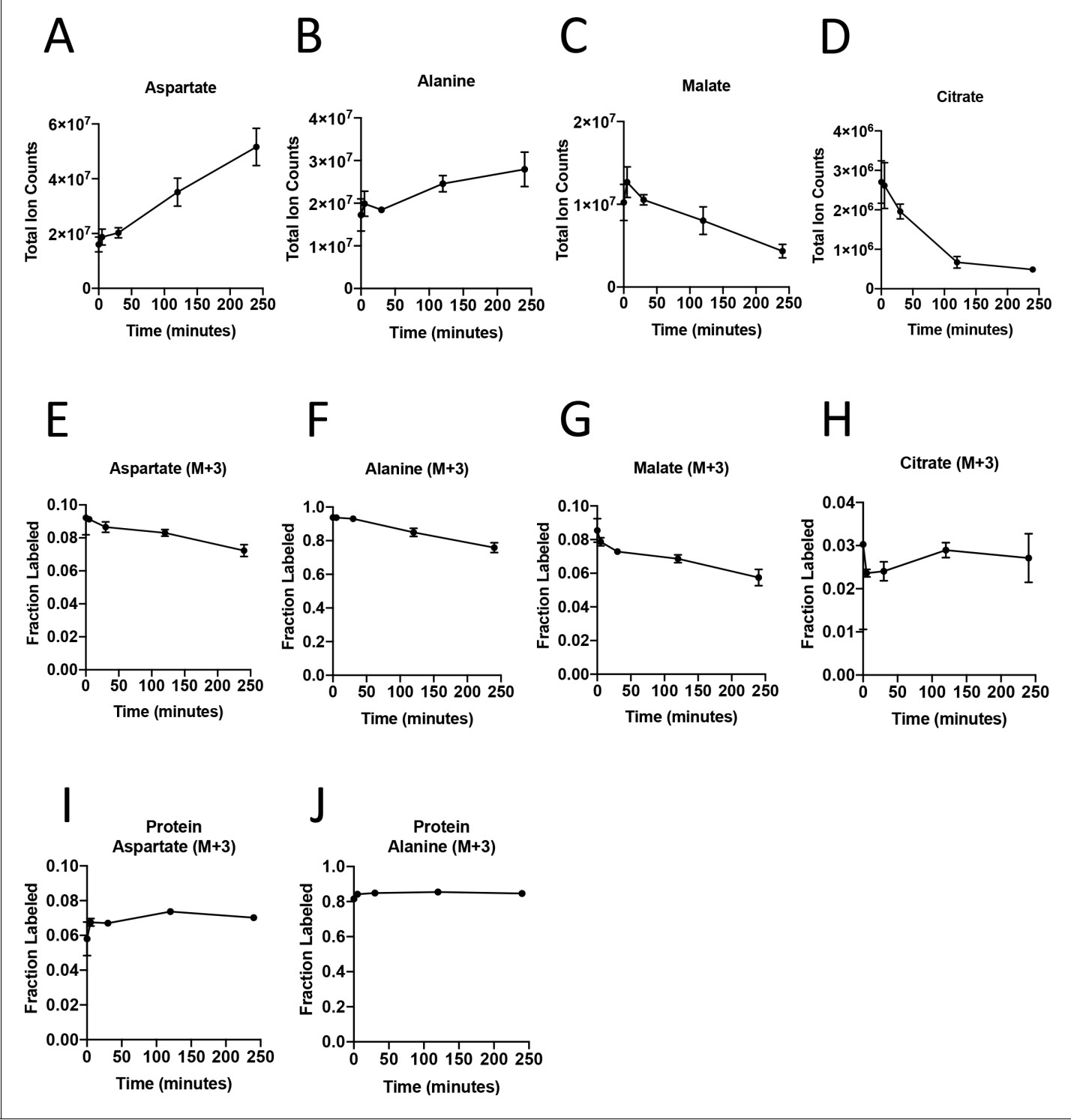

**Figure 3.** Turnover of polar metabolites but not protein is fast relative to the time needed to sort cells. (**A–D**) Metabolite levels (total ion counts, TIC) of (**A**) aspartate, (**B**) alanine, (**C**) malate, and (**D**) citrate were measured by GC-MS in AL1376 PDAC cells extracted 0–240 min after incubation in PBS on ice to mimic optimal sorting conditions. Mean +/- SD is shown. (**E–H**) Fractional labeling of (**E**) aspartate, (**F**) alanine, (**G**) malate, and (**H**) citrate were measured by GC-MS from cells extracted 0–240 min after incubation in PBS on ice. The M+3 isotopomer is shown. Mean +/- SD is shown. (**I–J**) Fractional labeling of (**I**) aspartate and (**J**) alanine from protein hydrolysates were measured by GC-MS in cells extracted 0–240 min after incubation in PBS on ice. The M+3 isotopomer is shown. Mean +/- SD is shown. Time points analyzed were 0 min (no PBS incubation), 5 min, 30 min, 120 min, or 240 min of incubation in PBS.

*Figure 3 continued on next page*

*Figure 3 continued*

The online version of this article includes the following source data and figure supplement(s) for figure 3:

**Source data 1.** Metabolite abundance, metabolite isotope labeling, and protein hydrolysate isotope labeling by U-$^{13}$C- glucose in AL1376 PDAC cells.
**Figure supplement 1.** Turnover of polar metabolites but not protein is fast relative to the time needed to sort cells.

*figure supplement 1C*, *Figure 4—figure supplement 1—source data 1*), matching the lower fractional labeling we observed from glucose. We did not observe other appreciable differences in fractional labeling of glutamate or proline from glutamine in protein between cancer cells and PSCs (*Figure 4—figure supplement 1D–E*, *Figure 4—figure supplement 1—source data 1*). Taken together, these data suggest a differential fate for glucose in these cell types, with increased M+3 labeling of aspartate from glucose carbon in cancer cells relative to PSCs.

Because the labeling of amino acids in protein is unlikely to reach steady-state even after multiple days of labeling, one explanation for the difference in aspartate labeling from labeled glucose in the cancer cells relative to the PSCs is a higher rate of protein synthesis in the cancer cells, although this is unlikely to differentially affect only M+3 labeled species. Nevertheless, to examine this possibility, we measured protein synthesis rates in each cell type using a fluorescent protein synthesis reporter in which BFP is fused to an unstable *E. coli* dihydrofolate reductase (DHFR) domain. Upon addition of the DHFR active site ligand trimethoprim (TMP), the reporter is stabilized and the rate of fluorescence accumulation reflects the synthesis rate of the fluorescent protein (*Han et al., 2014*). Consistent with previous reports, this reporter produced similar results compared to an assessment of protein synthesis through incorporation rates of the aminoacyl tRNA analog puromycin (*Darnell et al., 2018*) when BFP accumulation after TMP addition was assayed over time in PDAC cancer cell and PSC mono-cultures and compared to cells with no TMP added as a negative control (*Figure 4—figure supplement 1F–G*). The BFP reporter is suitable for use in sorted cells from a mixed cell system, and thus was used to assess protein synthesis rates in cancer cells and PSCs in organoid co-cultures. Interestingly, even though cancer cells and PSCs exhibited a similar protein synthesis rate in monoculture (*Figure 4—figure supplement 1F–G*), accumulation of BFP fluorescence was slower in cancer cells compared to PSCs in 3D co-cultures (*Figure 4H*). This argues that protein synthesis rates are higher in PSCs in organoid co-cultures, and that the higher M+3 aspartate labeling observed in cancer cells cannot be explained by a higher rate of protein synthesis in the cancer cells in this co-culture system.

M+3 labeling from U-$^{13}$C-glucose is often used as a surrogate for pyruvate carboxylation activity, but can also occur from multiple rounds of TCA cycling (*Alves et al., 2015*). To more directly assess pyruvate carboxylation activity in additional experiments with multiple replicates, we traced 1-$^{13}$C-pyruvate or 3,4-$^{13}$C-glucose fate in organoid-PSC co-cultures. 1-$^{13}$C-pyruvate or 3,4-$^{13}$C-glucose can only label aspartate via pyruvate carboxylation, because the $^{13}$C-label is lost as carbon dioxide if pyruvate is metabolized to acetyl-coA via PDH prior to entering the TCA cycle (*Figure 4I*). Compared to U-$^{13}$C-glucose labeling, a greater difference and significantly higher M+1 aspartate labeling in protein was observed using 1-$^{13}$C-pyruvate or 3,4-$^{13}$C-glucose in sorted cancer cells compared to PSCs from organoid-PSC co-cultures, further supporting that pyruvate carboxylation activity is higher in these cells (*Figure 4J–K*, *Figure 4—figure supplement 1H*). Taken together, these data argue that cancer cells within PDAC organoid-PSC co-cultures have higher pyruvate carboxylation activity than PSCs.

To investigate whether PDAC cancer cells also exhibit higher pyruvate carboxylation activity in tumors in vivo, we first verified that the tdTomato fluorescence in tumors arising in KP$^{-/-}$C mice bearing a *LSL-tdTomato* allele did not co-localize with staining for the fibroblast-specific marker alpha-smooth muscle actin (α-SMA) (*Figure 5A*), but did co-localize with Cytokeratin 19 (CK19), a marker of pancreatic cancer cells (*Figure 5B*). This verifies that tdTomato labeling can be used to isolate cancer cells from α-SMA-positive fibroblasts, and a combination of tdTomato fluorescence and an antibody for the pan-hematopoietic marker CD45 allowed efficient sorting of cancer cells, fibroblasts, and hematopoietic cells as verified by expression of relevant mRNAs using qPCR (*Figure 5C–F*, *Figure 5—figure supplement 1A–B*). To label protein in PDAC tumors in vivo, autochthonous tumor-bearing mice were infused with U-$^{13}$C-glucose for 24 hr (*Figure 5—figure supplement 1C–D*), with aspartate labeling observed in protein hydrolysates from bulk tumors in this time frame

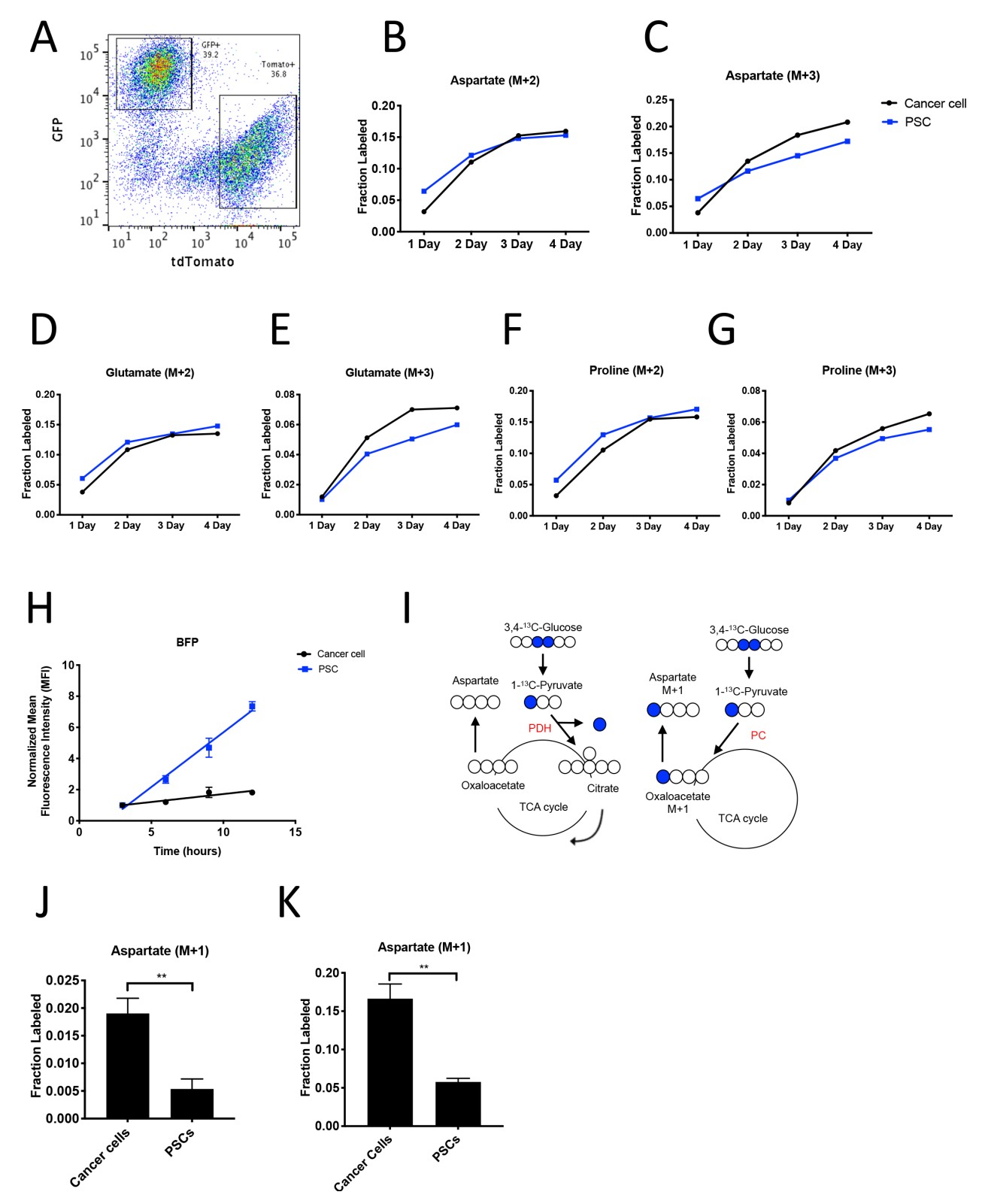

**Figure 4.** PDAC cancer cells have higher pyruvate carboxylation activity than fibroblasts in organoid co-cultures. (**A**) A representative flow cytometry plot showing GFP and tdTomato expression in PSCs and cancer cells respectively from digested organoid-PSC co-cultures. Cells were gated on the single cell, live population. (**B–G**) Fractional labeling of aspartate (**B–C**), glutamate (**D–E**), and proline (**F–G**) in protein hydrolysates from sorted organoid cancer cells (black) and PSCs (blue) after exposure of organoid co-cultures to U-$^{13}$C-glucose for the indicated time. Cancer cells were derived from *LSL-*

*Figure 4 continued on next page*

*Figure 4 continued*

Kras$^{G12D/+}$; Trp53$^{fl/fl}$; Pdx1-Cre; LSL-tdTomato (KP$^{-/-}$CT) mice. M+2 and M+3 isotopomers are shown as indicated. Each time point represents pooled organoid co-culture samples from one 24-well plate in a representative experiment. (H) BFP fluorescence as measured by flow cytometry of a protein synthesis reporter in the indicated cell type isolated from KP$^{-/-}$CT PDAC organoid-PSC co-cultures over time following TMP administration. BFP fluorescence is shown for cancer cells (black) and PSCs (blue). Mean +/- SD is shown. Data were normalized to no TMP controls. (I) Schematic illustrating the $^{13}$C labeling patterns for 3,4-$^{13}$C-glucose or 1-$^{13}$C-pyruvate tracing. TCA cycle metabolites and aspartate are unlabeled from these tracers if metabolized via PDH (left), but result in one carbon labeling (M+1) if metabolized via PC (right). (J) Fractional labeling of aspartate M+1 from protein hydrolysates following three days of KP$^{-/-}$CT organoid-PSC co-culture with 1-$^{13}$C-pyruvate. The difference in aspartate M+1 labeling was significant (p=0.0020) using an unpaired student's t test. Mean +/- SD is shown. (K) Fractional labeling of aspartate M+1 from protein hydrolysates following 3 days of KP$^{-/-}$CT organoid-PSC co-culture with 3,4-$^{13}$C-glucose. The difference in aspartate M+1 labeling was significant (p=0.0007) using an unpaired student's t test. Mean +/- SD is shown.

The online version of this article includes the following source data and figure supplement(s) for figure 4:

**Source data 1.** Isotope labeling of protein hydrolysates by U-$^{13}$C- glucose in organoid-PSC co-cultures after sorting.

**Figure supplement 1.** PDAC cancer cells have higher pyruvate carboxylation activity than fibroblasts in organoid co-cultures.

**Figure supplement 1—source data 1.** Isotope labeling of protein hydrolysates by U-$^{13}$C- glutamine in organoid-PSC co-cultures after sorting.

(*Figure 5G*). Cell populations were sorted from tumors, and labeling of amino acids was determined in protein hydrolysates from each cell population as well as from protein obtained from the bulk digested tumor (unsorted). In agreement with labeling patterns from organoid-co-cultures, tdTomato$^+$ cancer cells from PDAC tumors in mice had the highest M+3 protein aspartate labeling in protein, as well as higher M+2 aspartate labeling (*Figure 5H–I*, *Figure 5—source data 1*). This labeling pattern was also reflected in higher M+2 and M+3 labeling in glutamate but not in other glucose-labeled amino acids in protein (*Figure 5—figure supplement 1E–J*, *Figure 5—source data 1*). Taken together, these data are consistent with increased pyruvate carboxylation, as well as increased glucose oxidation via PDH, in the cancer cells relative to the stromal cell populations analyzed in PDAC tumors in vivo.

## Pyruvate carboxylase expression in cancer cells is required for PDAC tumor growth in vivo

To test whether PC is responsible for the observed pyruvate carboxylation activity and is functionally important for cells to proliferate in organoid co-cultures and tumors, PC expression was disrupted in murine PDAC cell lines, organoids, and PSCs using CRISPR/Cas9. First, CRISPRi was used to knock down PC expression in a PDAC cancer cell line derived from KP$^{-/-}$C mice (*Figure 6—figure supplement 1A*; *Horlbeck et al., 2016*). The ratio of M+1 aspartate to M+1 pyruvate derived from 1-$^{13}$C-pyruvate or 3,4-$^{13}$C-glucose has been used as a way to approximate pyruvate carboxylation activity (*Davidson et al., 2016*). As expected, PC knockdown in these PDAC cells resulted in a decrease in aspartate labeling from 1-$^{13}$C-pyruvate and relative pyruvate carboxylation activity compared to control cells as assessed by the ratio of M+1 labeled aspartate to M+1 labeled pyruvate (*Figure 6—figure supplement 1B–C*), but did not affect proliferation in culture (*Figure 6—figure supplement 1D*). However, knockdown of PC in PDAC organoids reduced growth of these cells in organoid-PSC co-cultures (*Figure 6—figure supplement 1E–G*). PC expression level and aspartate labeling from 1-$^{13}$C-pyruvate were increased by exogenous PC expression in PDAC PC knockdown cells (*Figure 6—figure supplement 1H–J*). When transplanted subcutaneously, PDAC cell lines with PC knockdown formed tumors that grew similarly to control cells (*Figure 6—figure supplement 1K*); however, PC expression was similar or increased in the tumors formed from PC knockdown cells compared to control tumors (*Figure 6—figure supplement 1L*). These data suggest that over time, cells that grew into tumors were selected for reversal of PC knockdown and that PC is required for PDAC tumor growth in vivo even though it is dispensable in culture, as has been observed previously in lung cancer (*Davidson et al., 2016*; *Fan et al., 2009*; *Sellers et al., 2015*).

To further test the requirement for PC in PDAC tumors, we generated cancer cell clones with complete CRISPR/Cas9 disruption of PC expression (*Figure 6—figure supplement 1M–N*). Similar to knockdown experiments, loss of PC had no effect on proliferation of PDAC cells in culture (*Figure 6A*), whereas loss of PC reduced the growth of organoid co-cultures (*Figure 6B–C*). CRISPR/Cas9 was also used to knockout PC in PSCs, and despite loss of PC expression and reduced pyruvate carboxylation activity (*Figure 6—figure supplement 2A–C*), PC knockout PSCs retained the

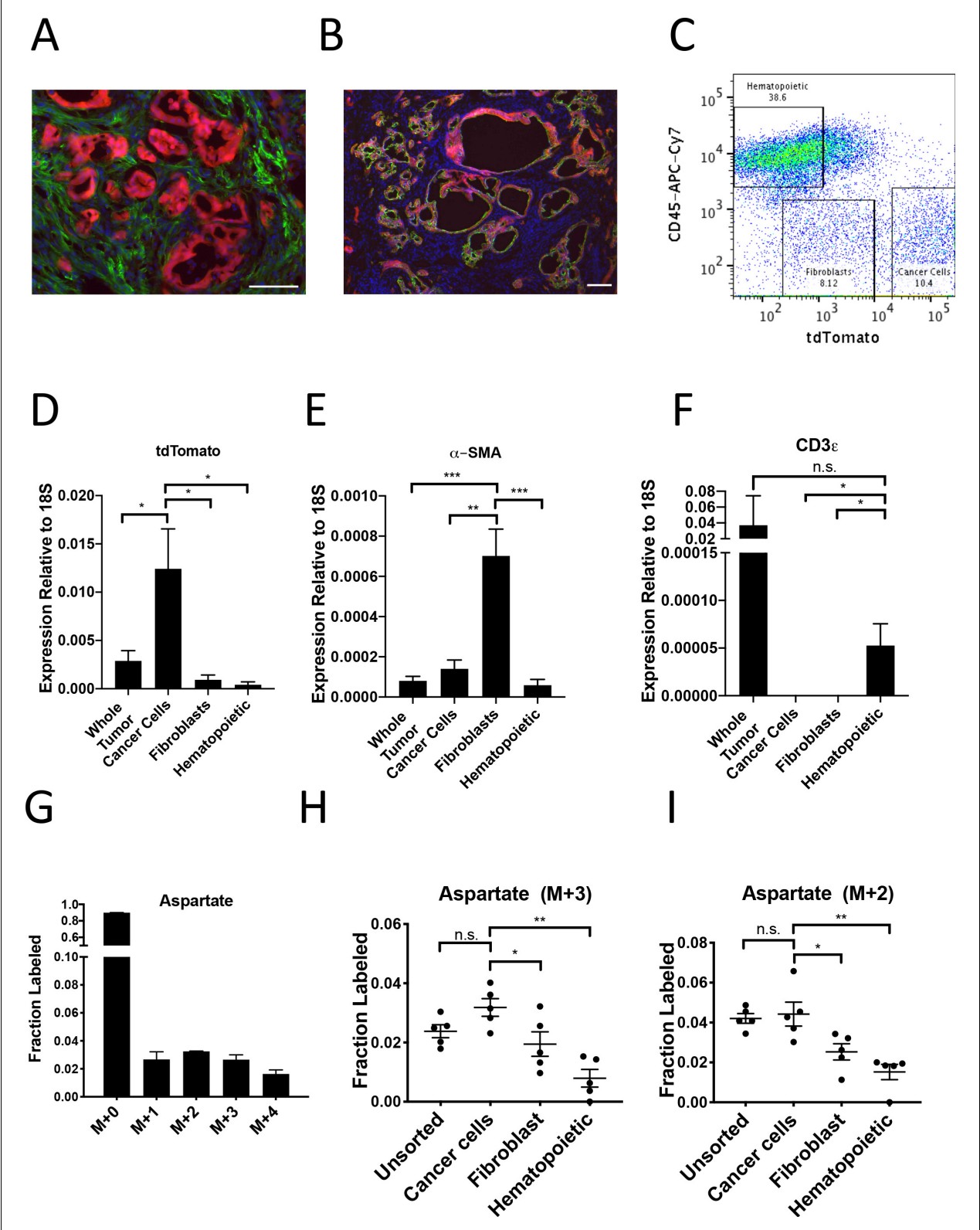

**Figure 5.** PDAC cancer cells have higher pyruvate carboxylation activity in vivo. (**A–B**) Tumors from *LSL-Kras*^*G12D/+*^*; Trp53*^*fl/fl*^*; Pdx1-Cre; LSL-tdTomato* (KP^-/-^CT) mice were stained with antibodies against RFP (red) and (**A**) α-SMA (green), a fibroblast marker, or (**B**) CK19 (green), a cancer cell marker. Scale bars represent 25 µm. (**C**) A representative flow cytometry plot showing CD45 and tdTomato expression in cells derived from a PDAC tumor arising in KP^-/-^CT mice. Cells were gated on the live population. (**D**) Expression of tdTomato was measured by qPCR in sorted cells from KP^-/-^CT PDAC tumors.
*Figure 5 continued on next page*

*Figure 5 continued*

The difference in expression between cancer cells and whole tumor (p=0.0430), fibroblasts (p=0.0156), or hematopoietic cells (p=0.0119) was significant based on unpaired, two-tailed student's t-tests. Mean +/- SEM is shown. n = 6 mice. 18S was used as a housekeeping gene control. (E) Expression of α-SMA was measured by qPCR in sorted cells from KP$^{-/-}$CT PDAC tumors. The difference in expression between fibroblasts and whole tumor (p=0.0004), cancer cells (p=0.0013), or hematopoietic cells (p=0.0003) was significant based on unpaired, two-tailed student's t-tests. Mean +/- SEM is shown. n = 6 mice. 18S was used as a housekeeping gene control. (F) Expression of CD3ε was measured by qPCR in sorted cells from KP$^{-/-}$CT PDAC tumors. The difference in expression between hematopoietic cells and whole tumor was not significant (p=0.3347), but the difference between hematopoietic cells and cancer cells (p=0.0378) or fibroblasts (p=0.0379) was significant based on unpaired, two-tailed student's t-tests. Mean +/- SEM is shown. n = 6 mice. 18S was used as a housekeeping gene control. (G) Fractional labeling of aspartate from protein hydrolysates from intact PDAC tumors following a 24 hr U-$^{13}$C-glucose infusion at a rate of 30 mg/kg/min. n = 4. Mean +/- SEM is shown. (H) Fractional labeling of aspartate from protein hydrolysates from the indicated sorted cell populations from tumors arising in KP$^{-/-}$CT mice following a 24 hr U-$^{13}$C-glucose infusion at a rate of 30 mg/kg/min. The M+3 isotopomers are shown. The differences in M+3 aspartate labeling in cancer cells compared to fibroblasts (p=0.0413) and hematopoietic cells (p=0.0005) were significant, and the difference between cancer cells and unsorted tumor cells (p=0.0612) was not significant based on unpaired, two-tailed student's t-tests. Mean +/- SEM is shown. n = 5 mice. (I) Fractional labeling of aspartate from protein hydrolysates from the indicated sorted cell populations from tumors arising in KP$^{-/-}$CT mice following a 24 hr U-$^{13}$C-glucose infusion at a rate of 30 mg/kg/min. The M+2 isotopomers are shown. The differences in M+2 aspartate labeling in cancer cells compared to fibroblasts (p=0.0309) and hematopoietic cells (p=0.0035) were significant, and the difference between cancer cells and unsorted tumor cells (p=0.7444) was not significant based on unpaired, two-tailed student's t-tests. Mean +/-SEM is shown. n = 5 mice.

The online version of this article includes the following source data and figure supplement(s) for figure 5:

**Source data 1.** Isotope labeling of protein hydrolysates in mice with autochthonous PDAC tumors after 24 hr of U-$^{13}$C- glucose infusion and sorting.

**Figure supplement 1.** PDAC cancer cells have higher pyruvate carboxylation activity in vivo.

ability to enhance PDAC organoid growth or growth of PDAC cancer cells as tumors in subcutaneous transplants, although the effect was reduced compared to sgControl PSCs (*Figure 6—figure supplement 2D–F*). Consistent with a requirement for PC expression in cancer cells to form PDAC tumors, PC-null cancer cells did not form tumors when transplanted into syngeneic mice subcutaneously or orthotopically (*Figure 6D–E*). However, surprisingly, PC-null cancer cells still displayed M+1 aspartate labeling from 1-$^{13}$C-pyruvate with similar or only a slight decrease in pyruvate carboxylation activity compared to control cells (*Figure 6F–G*). Taken together, these data argue that loss of PC in cancer cells can impact tumor growth, but another enzyme must also contribute to pyruvate carboxylation activity in these cells.

## Reversible malic enzyme 1 activity contributes to pyruvate carboxylation activity in PDAC cells and is important for tumor growth in vivo

A candidate for the pyruvate carboxylation activity observed in PC-null cells is malic enzyme, since this enzyme catalyzes the interconversion of pyruvate and $CO_2$ with malate, another 4-carbon TCA cycle intermediate. Malic enzyme is typically assumed to catalyze malate decarboxylation as a source of NADPH in cells (*Cairns et al., 2011*; *Hosios and Vander Heiden, 2018*), but has previously been shown to be reversible and produce malate from pyruvate and $CO_2$ in purified enzyme assays (*Ochoa et al., 1947*; *Ochoa et al., 1948*). Thus, we tested whether malic enzyme activity could sustain M+1 labeling of aspartate in PDAC cancer cells lacking PC by using CRISPR/Cas9 to knock out malic enzyme 1 (ME1). After knockout of both PC and ME1 in PDAC cell lines, aspartate labeling from 1-$^{13}$C-pyruvate is virtually abolished, suggesting that ME1 activity can contribute to pyruvate carboxylation activity in these cells (*Figure 7A–B*, *Figure 7—figure supplement 1A*). This aspartate labeling was also increased after exogenous expression of ME1 in PC and ME1 double knockout cells (*Figure 7A–B*). We used CRISPR/Cas9 to knockout or knockdown both PC and ME1 in organoids, which also resulted in decreased M+1 aspartate labeling from 1-$^{13}$C-pyruvate and decreased pyruvate carboxylation activity (*Figure 7—figure supplement 1B–D*). We also used CRISPR/Cas9 to knockout ME1 alone in PDAC cell lines and organoids (*Figure 7—figure supplement 1E–H*). Reduction or loss of ME1 alone in PDAC cell lines resulted in lower aspartate M+1 labeling and pyruvate carboxylation activity from 1-$^{13}$C-pyruvate (*Figure 7—figure supplement 1F–K*), further suggesting a role for ME1 in anaplerosis. This aspartate labeling was also increased after exogenous expression of ME1 in ME1 knockdown cells (*Figure 7—figure supplement 1L–N*). Interestingly, ME1 expression in KP$^{-/-}$C mouse and human PDAC tumors and organoids mimics that of PC in that it is more highly

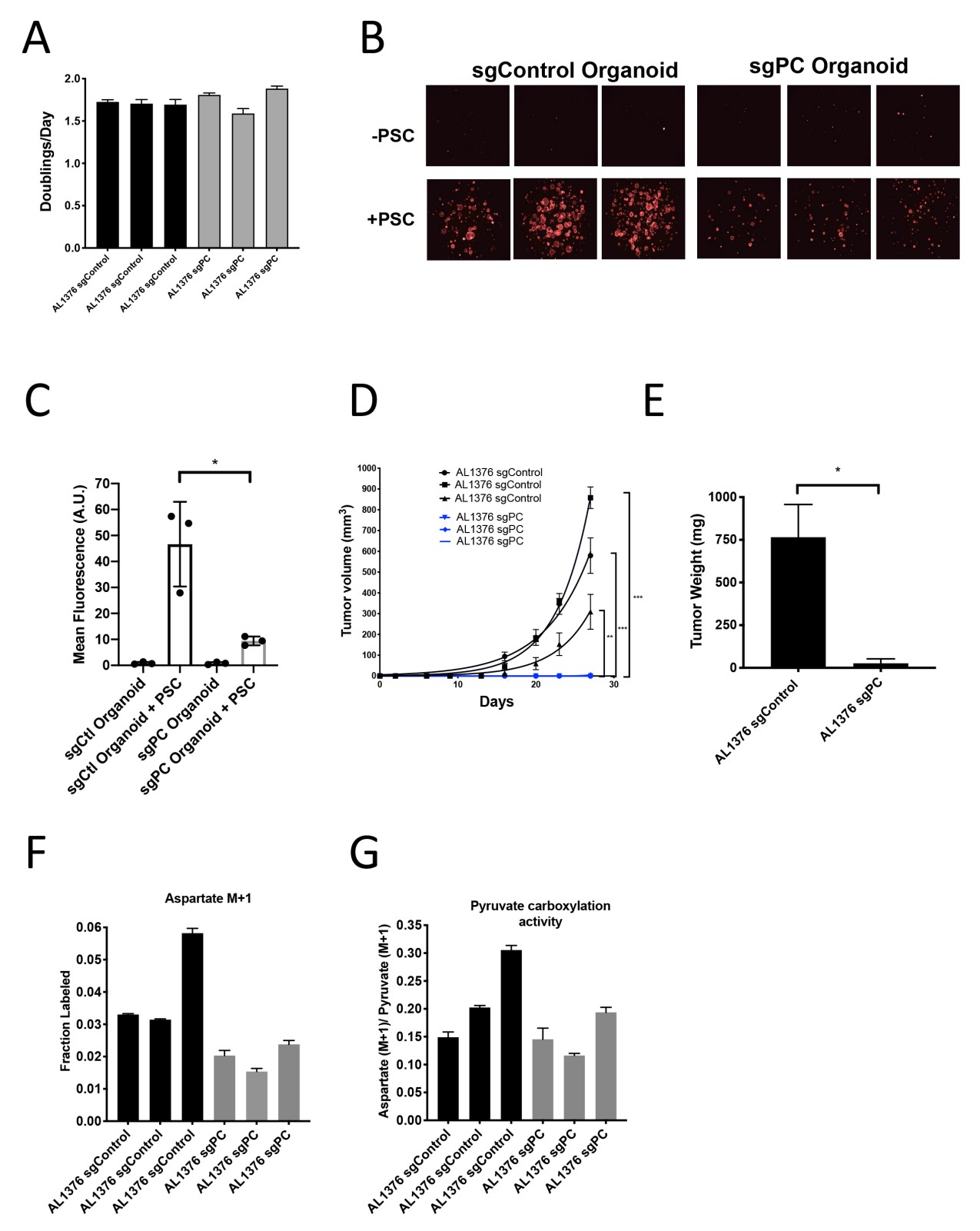

**Figure 6.** Pyruvate carboxylase in cancer cells is required for PDAC tumor growth in vivo. (**A**) Proliferation rate of AL1376 murine PDAC cells without (sgControl) or with (sgPC) deletion in standard 2D culture. Mean +/- SD is shown. (**B**) Fluorescent images of murine PDAC cancer cell organoids expressing tdTomato without (sgControl) or with (sgPC) deletion cultured in DMEM-pyruvate with 10% dialyzed FBS alone (top) or with murine PSCs (bottom). (**C**) Quantification of tdTomato fluorescence from images in (**B**). Control organoids with PSCs had significantly higher tdTomato fluorescence

*Figure 6 continued on next page*

*Figure 6 continued*

than sgPC organoids grown with PSCs (p=0.0171) based on an unpaired, two-tailed student's t-test. Mean +/- SD is shown. (**D**) Growth of sgControl (black) and sgPC (blue) AL1376 murine PDAC cells as tumors following subcutaneous transplantation into syngeneic Bl6 mice. The final tumor volume is significantly greater in sgControl AL1376 cells compared to sgPC cells based on unpaired, two-tailed student's t-tests (p<0.0001 to 0.0049). Mean +/- SEM is shown. n = 6 for each group. (**E**) Growth of sgControl and sgPC AL1376 murine PDAC cells as tumors after orthotopic transplantation into the pancreas of syngeneic Bl6 mice. Tumor weight was measured after 21 days and was significantly greater in mice transplanted with sgControl cells based on an unpaired, two-tailed student's t-test (p=0.0050). Mean +/- SEM is shown. n = 5 mice for each group. (**F**) Fractional labeling of aspartate in sgControl and sgPC AL1376 murine PDAC cells after 24 hr of culture with 1-$^{13}$C-pyruvate. Mean +/- SD is shown. (**G**) Aspartate M+1 isotopomer labeling from (**F**) was normalized to pyruvate M+1 labeling as a surrogate for pyruvate carboxylation activity. Mean +/- SD is shown.

The online version of this article includes the following figure supplement(s) for figure 6:

**Figure supplement 1.** Pyruvate carboxylase in cancer cells is required for PDAC tumor growth in vivo.
**Figure supplement 2.** Pyruvate carboxylase knockout PSCs retain ability to enhance PDAC growth.

expressed in cancer cells compared to stroma, suggesting that ME1 could also contribute to the higher pyruvate carboxylation seen in cancer cells compared to PSCs (*Figure 7C–E*, *Figure 7—figure supplement 2A–E*).

We next assessed whether ME1 was essential for tumor and organoid growth. Similar to loss of PC, loss of ME1 had minimal effect on cancer cell proliferation in monoculture (*Figure 7F*, *Figure 7—figure supplement 1F*), but reduced growth of organoid co-cultures compared to controls (*Figure 7G–H*, *Figure 7—figure supplement 1E*). Transplantation of ME1-null cancer cells in vivo also resulted in reduced tumor growth (*Figure 7I*), consistent with published data (*Son et al., 2013*). Taken together, these data argue that both PC and ME1 are important enzymes for PDAC cancer cells in tumors and can contribute to the pyruvate carboxylation activity observed in pancreatic cancer.

## Discussion

Metabolism can differ between cancer cells in culture and tumors, and understanding how nutrients are used by cancer cells in vivo has been an area of interest for developing cancer therapies. Tumor metabolic phenotypes have been assumed to reflect the metabolism of cancer cells within a tumor; however, in many tumors such as in PDAC, cancer cells are a minority cell population. Metabolic interactions between cell types have been described in normal tissues (*Bélanger et al., 2011*), and some metabolic phenotypes observed in cancer cells such as increased glucose utilization are also prominent in other cell types including fibroblasts and immune cells that can be abundant in some tumors (*Lemons et al., 2010*; *Vincent et al., 2008*; *Zhao et al., 2019*). Therefore, methods to deconvolute which cell types in a tumor are responsible for observed tissue metabolic phenotypes are needed.

We find that pancreatic tumors exhibit evidence of glucose metabolism, with carboxylation of glucose-derived pyruvate being more active in cancer cells than in other tumor cell types. However, because glucose will label both pyruvate and lactate, and these nutrients can be exchanged between cell types, it cannot be concluded that the cancer cells necessarily derive TCA cycle metabolites directly from glucose in a cell autonomous manner. In fact, rapid exchange of labeled intracellular and extracellular pyruvate and lactate among cell types is likely, making it difficult to address the original cellular source of labeled TCA metabolites with these methods. Thus, while this approach could be used to understand differential pathway use between cell types, in many cases it will not be able to distinguish the exact source of carbon that labels metabolites in individual cells.

Another important caveat to interpreting labeling patterns in protein or other macromolecules in cells within tissues is that labeling is unlikely to reach steady state, particularly for analysis of cells in tissues in vivo. This failure to reach steady state means that differences in label delivery or uptake could cause differences in biomass labeling even when the pathway involved in labeling is similarly active in both cell types. Thus, controlling for variables such as biomass synthesis rates between cell types can help with data interpretation. The ability to reach a pseudo-metabolic steady state facilitates interpretation of labeling data; however, this requires that both circulating nutrient levels and labeling patterns are relatively constant (*Buescher et al., 2015*; *Jang et al., 2018*). Glucose infusion rates and techniques can vary across studies (*Davidson et al., 2016*; *Faubert et al., 2017*;

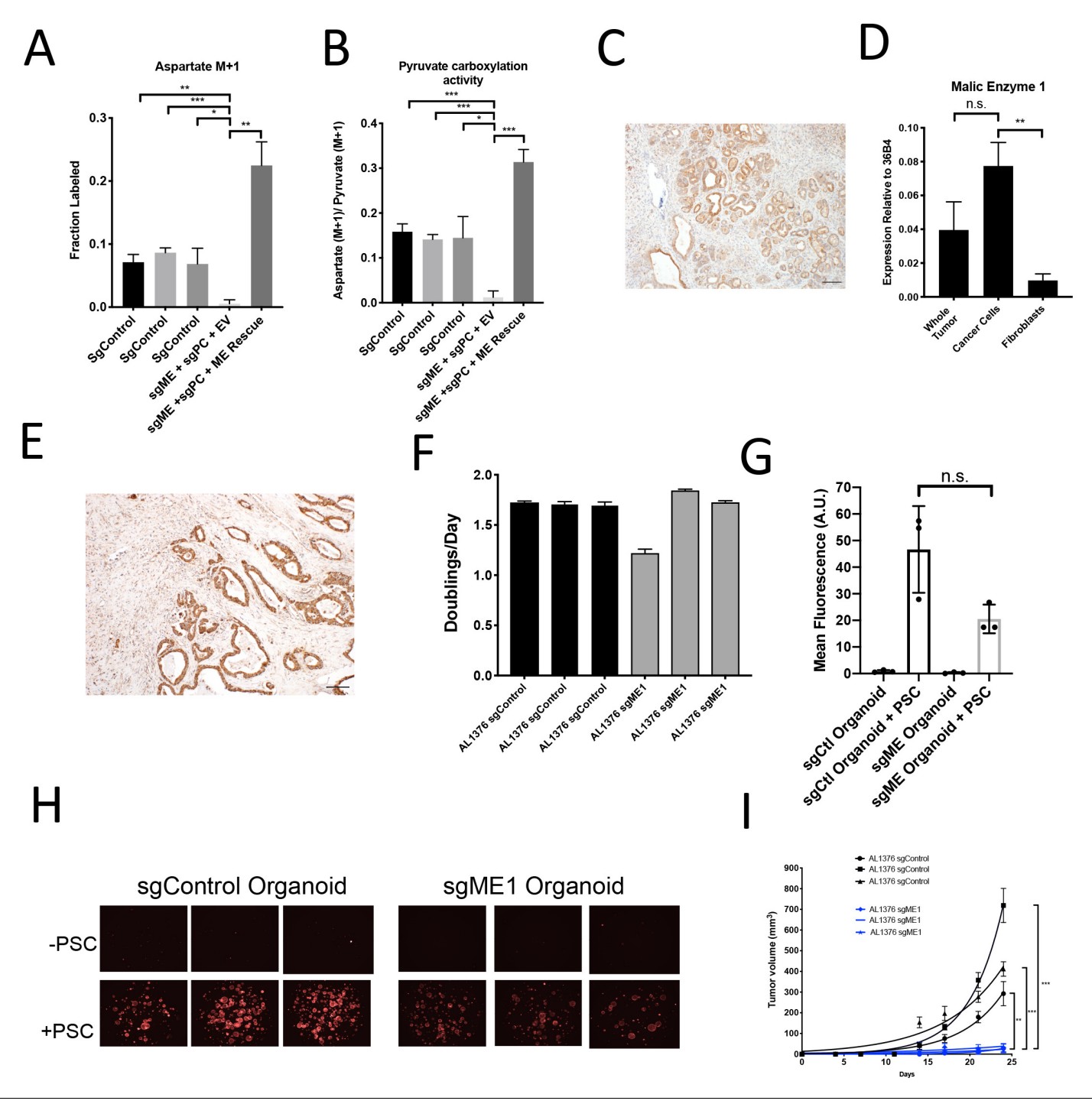

**Figure 7.** Malic enzyme 1 contributes to pyruvate carboxylation activity in PDAC cells and is important for tumor growth. (A–B) CRISPR/Cas9 was used to disrupt PC and/or ME1 as indicated in AL1376 murine PDAC cells. (A) Fractional labeling of M+1 aspartate following culture of the indicated cells for 24 hr in media containing 1-$^{13}$C-pyruvate. M+1 aspartate labeling was significantly decreased in double knockout cells compared to control cells (p=0.0012, 0.0001, and 0.0131) or to double knockout cells with exogenous ME1 expression (ME rescue) (p=0.0006) based on unpaired, two-tailed student's t-tests. Mean +/- SD is shown. (B) Aspartate M+1 isotopomer labeling was normalized to pyruvate M+1 labeling as a surrogate for pyruvate carboxylation activity following 24 hr of 1-$^{13}$C-pyruvate tracing. Pyruvate carboxylation activity was significantly decreased in double knockout cells compared to control cell lines (p=0.0004, 0.0002, and 0.0100) or to double knockout cells with exogenous ME1 expression (ME rescue) (p<0.0001) based on unpaired, two-tailed student's t-tests. Mean +/- SD is shown. (C) Sections from tumors arising in *LSL-Kras*$^{G12D/+}$; *Trp53*$^{fl/fl}$; *Pdx1-Cre* (KP$^{-/-}$C) mice were stained with an antibody against ME1. Scale bar represents 100 μm. (D) Expression of ME1 was measured by qPCR in the indicated cells sorted from tumors arising in KP$^{-/-}$C mice. The expression of ME1 was not significantly different in sorted cancer cells compared to the whole tumor
*Figure 7 continued on next page*

*Figure 7 continued*

(p=0.1114), but was significantly higher in sorted cancer cells compared to fibroblasts (p=0.0009) based on unpaired, two-tailed student's t-tests. Mean +/- SEM is shown. 36B4 was used as a housekeeping gene control. (E) Representative image from a human pancreatic tumor tissue microarray stained with an antibody against ME1. Scale bar represents 100 μm. (F) Proliferation rate of sgControl and sgME1 AL1376 murine PDAC cells in standard 2D culture. (G) Quantification of tdTomato fluorescence from images of sgControl or sgME1 PDAC cancer cell organoids isolated from KP$^{-/-}$CT tumors cultured in DMEM-pyruvate with 10% dialyzed FBS alone or with murine PSCs. sgControl organoids with PSCs trended towards higher tdTomato fluorescence compared to sgME1 organoids with PSCs (p=0.0579) but was not significant based on an unpaired, two-tailed student's t-test. Mean +/- SD is shown. The sgControl data are also shown in *Figure 6B–C*. (H) Fluorescent images of sgControl or sgME1 PDAC cancer cell organoids cultured DMEM-pyruvate with 10% dialyzed FBS alone (top) or with murine PSCs (bottom). The sgControl images are also shown in *Figure 6B–C*. (I) Growth of sgControl (black) and sgME1 (blue) AL1376 murine PDAC cells as tumors following subcutaneous transplantation into syngeneic B6 mice. The final tumor volume is significantly greater in sgControl AL1376 cells compared to sgME1 cells based on unpaired, two-tailed student's t-tests (p<0.0001 to 0.0017). Mean +/- SEM is shown. n = 6 for each group.

The online version of this article includes the following figure supplement(s) for figure 7:

**Figure supplement 1.** Malic enzyme 1 contributes to pyruvate carboxylation activity in PDAC cells and is important for tumor growth.

**Figure supplement 2.** Expression of ME1 in a human PDAC tissue microarray and in murine organoids and stroma.

*Hui et al., 2017*; *Ma et al., 2019*; *Marin-Valencia et al., 2012*), and these differences may affect whether circulating nutrient levels are constant. While this may be one reason why differences in labeling were observed across PDAC models evaluated in this study, additional factors such as tumor initiation and growth rates, cells of origin, p53 status, and different composition of cancer and stromal cells are known to exist as well (*Rosenfeldt et al., 2013*; *Vennin et al., 2019*).

While we did not directly assess the effects of cell sorting on metabolite levels or labeling in this study, we observed that for some metabolites, their levels and/or labeling patterns are not stable over the time needed to sort cells. However, the labeling patterns of some metabolites were maintained despite changes in levels, and might still be used to derive information about cell-specific metabolism, particularly when appropriate controls and orthogonal evidence support the conclusions.

We find that whether cells are grown in 2D cultures, in 3D organoid co-culture with PSCs, or as orthotopic or subcutaneous tumors impacts whether pyruvate carboxylation is important for proliferation, with the organoid and tumor models showing a similar dependency on this activity. Tumor organoid-stromal co-cultures represent a tractable model for metabolic characterization, and thus may be useful for exploration of other symbiotic metabolic relationships between pancreatic cancer cells and fibroblasts. However, while the difference in M+3 aspartate labeling seen in vivo was recapitulated by organoid-fibroblast co-cultures, other differences such as higher M+2 aspartate and glutamate labeling observed in vivo were not observed in the co-culture model. Therefore, some aspects of cell type-specific metabolism are not recapitulated even in co-culture organoid systems.

We find that PC and ME1 expression in cancer cells are both important for PDAC tumor growth in vivo. A dependence on pyruvate carboxylation seems to be a characteristic of both PDAC and lung tumors in vivo that is not prominent in standard cell culture systems (*Christen et al., 2016*; *Davidson et al., 2016*; *Fan et al., 2009*; *Hensley et al., 2016*; *Sellers et al., 2015*). Why this is the case is not known, but PC is an important anaplerotic pathway for the TCA cycle, contributing to biosynthesis of macromolecules such as protein, nucleotides, and lipids in cancer cells. Glucose metabolism and increased glucose uptake have been shown to be important for biosynthesis in PDAC tumors (*Santana-Codina et al., 2018*; *Ying et al., 2012*), but it has also been suggested that some PDAC tumors rely less on glucose for fuel and instead on alternative nutrient sources such as circulating lactate and glutamine (*Hui et al., 2017*), or alanine from stromal autophagy (*Sousa et al., 2016*). We did not observe differences in protein alanine labeling from glucose in either PSCs or organoids, although it remains possible the cells differentially utilize alanine acquired from a source other than glucose. For example, macropinocytosis to catabolize extracellular protein can be an important source of amino acids for cells in PDAC tumors (*Commisso et al., 2013*; *Davidson et al., 2017*). Nevertheless, the findings that PDAC tumors are FDG-PET positive (*Nguyen et al., 2011*; *Parikh et al., 2015*; *Yeh et al., 2018*) and that levels of glucose are depleted in tumor interstitial fluid relative to plasma in PDAC mouse models (*Sullivan et al., 2019*), are consistent with glucose being consumed by at least some cell types within the tumor.

Glutamine is also a source of TCA anaplerotic carbon that may contribute to biosynthesis differentially between cancer cells and stroma. Previous work has suggested that utilization of ME1 to produce pyruvate from glutamine can be important for PDAC cells to maintain redox balance, specifically via NADPH generation in vitro (*Son et al., 2013*), and that glutamine can be a major contributor to TCA metabolites in PDAC tumors (*Hui et al., 2017*). A potential role for malic enzyme in pyruvate carboxylation suggests use of this enzyme to produce malate could be another pathway for TCA cycle anaplerosis. Of note, this reaction would require NADPH, and may be more favored in cancer cells that exhibit a reduced redox state (*Hosios and Vander Heiden, 2018*). Furthermore, other pathways produce NADPH in cancer cells, including the pentose phosphate pathway, the one-carbon cycle, or isocitrate dehydrogenase (*Chen et al., 2019*). We also considered phosphoenolpyruvate carboxykinase (PEPCK) or malic enzymes 2 and 3 as possible contributors to pyruvate carboxylation activity, although these reactions are less energetically favorable in the reverse direction in comparison to malic enzyme 1; malic enzyme 1 is cytosolic, which is thought to be a more reducing environment than the mitochondria where malic enzymes 2 and 3 are localized (*Hu et al., 2008*). We did not see evidence for differential glutamine utilization in our organoid-PSC co-cultures, and PDAC tumors are resistant to glutaminase inhibitors (*Biancur et al., 2017*), but further work is needed to assess how glutamine metabolism and other anaplerotic pathways might be differentially active in cancer cells and non-cancer cells in PDAC tumors.

In pancreatic β-cells, PC and ME are thought to be part of a coordinated metabolic cycle that regulates insulin secretion (*Pongratz et al., 2007*). In this pyruvate cycle, ME1 generates NADPH and produces pyruvate from malate in the cytosol, which can then be used by PC to generate oxaloacetate in the mitochondria (*Pongratz et al., 2007*). While loss of ME activity might be expected to impact pyruvate carboxylation activity when both enzymes are present, the fact that residual pyruvate carboxylation activity is observed in the absence of PC, and that this is lost upon ME1 disruption argues that PC and ME may have redundant metabolic functions under some conditions. Surprisingly, isotope labeling in cells with loss of ME1 alone showed a larger decrease in labeling consistent with pyruvate carboxylation than was observed with PC knocked out and ME1 left intact. However, these data should not be used to conclude that flux through ME1 is higher than PC, particularly in cells where both enzymes are expressed and pyruvate cycling can occur. PC and ME1 are each essential for pancreatic tumors in vivo, despite a possible redundancy in pyruvate carboxylation activity. This may be because pyruvate cycling is important for tumor growth, or the need for anaplerosis to support tumor growth is more constrained in tumors than in cell culture. Indeed, a dependence on pyruvate carboxylation seems to be a characteristic of tumors in vivo that is not observed in culture across many cancer models (*Christen et al., 2016*; *Davidson et al., 2016*; *Fan et al., 2009*; *Hensley et al., 2016*; *Sellers et al., 2015*).

PC has been targeted with antisense oligonucleotides (*Kumashiro et al., 2013*) and relatively non-specific chemical inhibitors (*Bahl et al., 1997*; *Zeczycki et al., 2010*); however, inhibiting PC may have deleterious effects on whole body metabolism by interfering with gluconeogenesis or glucose-stimulated insulin secretion. Whether malic enzyme can compensate sufficiently for PC inhibition in those tissues to allow therapeutic targeting, or if malic enzyme is a viable alternative target, remains to be determined. Nevertheless, our data suggest that stable isotope tracing into macromolecules can be utilized to deconvolute complex tracing patterns in mammalian tissues and identify increased pathway activity in a particular cell type. Understanding the metabolic similarities and differences between cancer cells and stroma within PDAC and other tumors will be important in further delineating cancer-specific dependencies.

## Materials and methods

**Key resources table**

| Reagent type (species) or resource | Designation | Source or reference | Identifiers | Additional information |
|---|---|---|---|---|
| Strain, strain background (*Mus musculus*) | C57Bl6/J | Jax | Cat# JAX:000664, RRID:IMSR_JAX:000664 | Used for cell line transplantation |

*Continued on next page*

*Continued*

| Reagent type (species) or resource | Designation | Source or reference | Identifiers | Additional information |
|---|---|---|---|---|
| Genetic Reagent (*M. musculus*) | *LSL-Kras^{G12D/+} Trp53^{flox/flox} Pdx-1-Cre LSL-tdTomato;* KP^{-/-}CT | *Bardeesy et al., 2006* PMID:16585505 | | Mice from a mixed 129/Sv and C57Bl6/J genetic background as well as pure C57Bl6/J mice were used. Both sexes of mice were included in experiments. |
| Genetic Reagent (*M. musculus*) | *LSL-Kras^{G12D/+} Trp53^{R172H/+} Pdx-1-Cre LSL-tdTomato;* KPCT | *Hingorani et al., 2005* PMID:15894267 | | Mice from a mixed 129/Sv and C57Bl6/J genetic background as well as pure C57Bl6/J mice were used. Both sexes of mice were included in experiments. |
| Genetic Reagent (*M. musculus*) | *β-actin-GFP* | Jax | Cat# JAX:006567, RRID:IMSR_JAX:006567 | PSCs were isolated from *β-actin-GFP* mice in a C57Bl6/J background. |
| Cell line (*M. musculus*) | PSC | *Danai et al., 2018* PMID:29925948 | | Isolated from *β-actin-GFP* mice in a C57Bl6/J background (Jax 006567). PSCs were immortalized with TERT and SV40 largeT after several passages. |
| Cell line (*M. musculus*) | AL1376 | *Sullivan et al., 2018* PMID:29941931 | | Isolated from KP^{-/-}CT mouse PDAC tumor in a C57Bl6/J background. |
| Cell line (*M. musculus*) | Organoid | This paper | | Isolated from KP^{-/-}CT mouse PDAC tumor in a C57Bl6/J background |
| Biological sample (*Homo sapiens*) | Human pancreatic cancer tissue microarray | Biomax | PA961e | |
| Antibody | α-SMA (mouse monoclonal) | Sigma | Cat# F3777, RRID:AB_476977 | IF (1:500) |
| Antibody | CK19 (Rabbit monoclonal) | Abcam | Cat# ab133496, RRID:AB_ 11155282 | IF (1:100) |
| Antibody | tdTomato (Rabbit polyclonal) | Rockland | Cat# 600-401-379, RRID:AB_2209751 | IF (1:500) |
| Antibody | Pyruvate Carboxylase (Rabbit polyclonal) | Santa Cruz | Cat# sc-67021, RRID:AB_2283532 | IHC (1:50) |
| Antibody | Pyruvate Carboxylase (Mouse monoclonal) | Santa Cruz | Cat# sc-271493, RRID:AB_10649369 | WB (1:100) |
| Antibody | Malic Enzyme 1 (Rabbit polyclonal) | Proteintech | Cat# 16619-1-AP, RRID:AB_2143821 | IHC (1:200) WB (1:250) |
| Antibody | β-actin (Rabbit monoclonal) | Cell Signaling Technologies | Cat# 8457, RRID:AB_10950489 | WB (1:10,000) |
| Antibody | CD16/CD32 (Rat monoclonal) | Thermo Fisher | Cat# 14-0161-82, RRID:AB_467133 | FACS (10uL) |

*Continued on next page*

*Continued*

| Reagent type (species) or resource | Designation | Source or reference | Identifiers | Additional information |
|---|---|---|---|---|
| Antibody | CD45-APC-Cy7 (Rat monoclonal) | BD | Cat# 557659, RRID:AB_396774 | FACS (1:100) |
| Antibody | Puromycin (Mouse monoclonal) | Sigma | MABE343 | WB (1:25,000) |
| Antibody | Vinculin (Mouse monoclonal) | Abcam | Cat# ab18058, RRID:AB_444215 | WB (1:1000) |
| Commercial Assay or Kit | Protein A Antibody Purification Kit | Sigma | PURE1A | |
| Commercial Assay or Kit | RNAqueous-Micro Total RNA Isolation Kit | Life Technologies | AM1931 | |
| Commercial Assay or Kit | iScript cDNA Synthesis Kit | Bio-Rad Laboratories | 1708890 | |
| Commercial Assay or Kit | Pierce BCA Protein Assay Kit | Pierce | 23225 | |
| Commercial Assay or Kit | Amaxa Basic Nucleofector Kit for Primary Mammalian Epithelial Cells | Amaxa | VPI-1005 | |
| Recombinant DNA Reagent | pUSPmNG (plasmid) | *Li et al., 2019* PMID:31694929 | | U6 sgRNA PGK with mNeonGreen to express sgRNAs for PC and ME1 double knockout cell lines |
| Recombinant DNA Reagent | LentiCRISPRv2 (plasmid) | *Sanjana et al., 2014* PMID:25075903 | | Lentiviral construct to create PC and ME1 knockout cells |
| Recombinant DNA Reagent | Modified LentiCRISPRv2 (plasmid) | *Horlbeck et al., 2016* PMID:27661255 | | Modified dCas9-KRAB fusion protein to create PC and ME1 knockdown cells |
| Recombinant DNA Reagent | pLV-Hygro-EFS (plasmid) | Vectorbuilder | | Custom lentiviral construct for re-expressing PC or ME1 cDNA under a CMV promoter |
| Peptide, recombinant protein | EGF | Thermo Fisher | PMG8041 | |
| Peptide, recombinant protein | FGF | Peprotech | 100-26 | |
| Peptide, recombinant protein | Gastrin I | TOCRIS | 3006 | |
| Peptide, recombinant protein | Noggin | Peprotech | 250-38 | |
| Chemical compound, drug | Collagenase I | Worthington Biochemical | LS004194 | |

*Continued on next page*

*Continued*

| Reagent type (species) or resource | Designation | Source or reference | Identifiers | Additional information |
|---|---|---|---|---|
| Chemical compound, drug | Collagenase XI | Sigma | C9407 | |
| Chemical compound, drug | Collagenase P | Sigma | 11213865001 | |
| Chemical compound, drug | Dispase II | Roche | 04942078001 | |
| Chemical compound, drug | TrypLE Express | Thermo Fisher | 12605-010 | |
| Chemical compound, drug | DNAse I Type II | Sigma | D4527 | |
| Chemical compound, drug | DNAse I Type IV | Sigma | D5025 | |
| Chemical compound, drug | GlutaMAX | Thermo Fisher | 35050 | |
| Chemical compound, drug | HEPES | Thermo Fisher | 15630 | |
| Chemical compound, drug | TGF-b inhibitor A-83-01 | TOCRIS | 2939 | |
| Chemical compound, drug | Rho Kinase Inhibitor Y-27632 | Sigma | Y0503 | |
| Chemical compound, drug | N-Acetylcysteine (NAC) | Sigma | A9165 | |
| Chemical compound, drug | Nicotinamide | Sigma | N0636 | |
| Chemical compound, drug | B-27 supplement | Thermo Fisher | 17504 | |
| Chemical compound, drug | Methoxamine (MOX) reagent | ThermoFisher | TS-45950 | |
| Chemical compound, drug | N–methyl–N–(tert–butyldimethylsilyl) trifluoroacetamide + 1% tert–Butyldimethylchlorosilane | Sigma | 375934 | |
| Chemical compound, drug | Pyridine | Sigma | 270407 | |
| Chemical compound, drug | Nycodenz | VWR | 100356-726 | |
| Chemical compound, drug | Sulforhodamine B | Sigma | 230162 | |

*Continued*

| Reagent type (species) or resource | Designation | Source or reference | Identifiers | Additional information |
|---|---|---|---|---|
| Chemical compound, drug | SYBR Green Master Mix | Sigma | L6544 | |
| Software, algorithm | Prism | GraphPad | | Statistical analysis and graphing |
| Software, algorithm | FlowJo | BD | | Flow cytometry data analysis |
| Other | SYTOX Red Dead Cell Stain | Life Technologies | S34859 | FACS (1:1000) |
| Other | Advanced DMEM/F12 | Thermo Fisher | 12634 | |
| Other | DMEM without pyruvate | Corning | 10-017-CV | |
| Other | GBSS | Sigma | G9779 | |
| Other | Growth factor reduced (GFR) matrigel | Corning | 356231 | |
| Other | Plastic coverslips | Thermo | 174985 | |
| Other | Flow cytometry staining buffer | Thermo Fisher | 00-4222-57 | |

## Mouse models

All animal studies were approved by the MIT Committee on Animal Care under protocol #0119-001-22. For autochthonous models, *LSL-Kras*$^{G12D/+}$; *Trp53*$^{flox/flox}$; *Pdx1-Cre*; *LSL-tdTomato* (KP$^{-/-}$CT) (*Bardeesy et al., 2006*) and *LSL-Kras*$^{G12D/+}$; *Trp53*$^{R172H/+}$; *Pdx1-Cre*; *LSL-tdTomato* (KPCT) (*Hingorani et al., 2005*), mice from a mixed 129/Sv and C57Bl6/J genetic background as well as pure C57Bl6/J mice were used. C57Bl6/J mice were used for allografts. Both sexes of mice were included in experiments. Animals were housed under a 12 hr light and 12 hr dark cycle, and cohoused with littermates with ad libitum access to water and food unless otherwise stated.

## Glucose infusion

Infusion of U-$^{13}$C-glucose (Cambridge Isotope Laboratories) was performed as previously described (*Davidson et al., 2016*). Surgery was performed to implant a catheter into the jugular vein of animals 3–4 days prior to infusion. For 4–6 hr infusions, mice were fasted for 4 hr prior to beginning the infusion. For 24 hr infusions, mice were not fasted prior to infusion. Infusions were performed in conscious, free-moving animals for 4 or 24 hr at a rate of 30 mg/kg/min. For 6 hr infusions, each animal, regardless of body weight, was infused with a fixed volume of 300 μl of a 500 mg/ml glucose solution over 6 hr, which is an infusion rate of 0.4 mg/min.

Tumors were either digested for FACS or rapidly frozen using a Biosqueezer (BioSpec Products) and stored at −80°C prior to metabolite extraction.

## Isotope labeling experiments

100,000 adherent cells were plated in six-well plates, or organoids and organoid-PSC co-cultures were plated on plastic coverslips (Thermo 174985) in 24-well plates. The following day, the cells were washed three times with PBS and then isotope-labeled media was added for the specified length of time (24–72 hr). For U-$^{13}$C-glucose or 3,4-$^{13}$C-glucose tracing, DMEM without glucose and pyruvate was used, supplemented with 25 mM U-$^{13}$C-glucose or 3,4-$^{13}$C-glucose, 10% dialyzed FBS, and penicillin-streptomycin. For 1-$^{13}$C-pyruvate tracing, DMEM with glucose and without pyruvate was used, adding 2 mM 1-$^{13}$C-pyruvate and supplementing with 10% dialyzed FBS and penicillin-streptomycin.

## Polar metabolite extraction

Adherent cells were washed once with ice-cold saline on ice and then extracted with a 5:3:5 ratio of ice-cold HPLC-grade methanol:water:chloroform. Mouse tissue or coverslips containing organoids and organoid-PSC co-cultures were washed once with saline prior to extraction. Tissue or matrigel domes containing the organoids and organoid-PSC co-cultures were then rapidly frozen using a Bio-squeezer (BioSpec Products) and stored at −80℃ prior to metabolite extraction. Snap frozen tissues or organoids were extracted with a 5:3:5 ratio of ice-cold HPLC-grade methanol:water:chloroform. For mouse plasma, 10 μL of plasma was extracted with 600 μL ice-cold methanol. All samples were vortexed for 10 min at 4℃ followed by centrifugation for 5 min at maximum speed on a tabletop centrifuge (~21,000 xg) at 4℃. An equal volume of the aqueous phase of each sample was then dried under nitrogen gas and frozen at −80℃ until analysis. For organoid samples, two rounds of extraction were done to eliminate excess protein from matrigel.

## Protein hydrolysis

Acid hydrolysis of protein was performed as described previously (*Mayers et al., 2016*; *Sullivan et al., 2018*). Frozen tissue or cell pellets were boiled for 24 hr at 100℃ in 500 μL (cell pellets) −1 mL (tissue) 6M HCl for amino acid analysis (Sigma 84429). 50 μL (tissue) −100 μL (cell pellets) of HCl solution was then dried under nitrogen gas while heating at 80℃. Dried hydrolysates were stored at −80℃ until derivatization.

## GC-MS analysis

Polar metabolites were analyzed as described previously (*Lewis et al., 2014*). Dried free metabolite extracts were dissolved in 16 μL methoxamine (MOX) reagent (ThermoFisher TS-45950) and incubated at 37℃ for 90 min followed by addition of 20 μL N–methyl–N–(tert–butyldimethylsilyl)trifluoroacetamide + 1% tert–Butyldimethylchlorosilane (Sigma 375934) and incubated at 60℃ for 1 hr. Dried protein hydrolysates were re-dissolved in 16 μL HPLC grade pyridine (Sigma 270407) prior to derivatization with 20 μL N–methyl–N–(tert–butyldimethylsilyl)trifluoroacetamide + 1% tert–Butyldimethylchlorosilane (Sigma 375934) at 60℃ for 1 hr. Following derivatization, samples were analyzed using a DB-35MS column (Agilent Technologies) in an Agilent 7890 gas chromatograph coupled to an Agilent 5975C mass spectrometer. Helium was used as the carrier gas at a flow rate of 1.2 mL/min. One microliter of sample was injected at 270℃. After injection, the GC oven was held at 100℃ for 1 min and increased to 300℃ at 3.5 ℃/min. The oven was then ramped to 320℃ at 20 ℃/min and held for 5 min. at this 320℃. The MS system operated under electron impact ionization at 70 eV and the MS source and quadrupole were held at 230℃ and 150℃, respectively. The detector was used in scanning mode, and the scanned ion range was 100–650 m/z. Data were corrected for natural isotope abundance.

## Adherent cell culture

Cell lines were cultured in DMEM (Corning 10–013-CV) supplemented with 10% fetal bovine serum and penicillin-streptomycin. Cell lines were regularly tested for mycoplasma contamination using the MycoAlert Plus kit (Lonza) or the Mycoprobe Mycoplasma Detection Kit (R and D Systems). PSCs were isolated from *β-actin-GFP* mice in a C57Bl6/J background (006567) as previously described (*Apte, 2011*; *Danai et al., 2018*): 3 mL of 1.3 mg/mL cold collagenase P (Sigma 11213865001) and 0.01 mg/mL DNAse (Sigma D5025) in GBSS (Sigma G9779) were injected into the pancreas. The tissue was then placed into 2 mL of collagenase P solution on ice. Cells were then placed in a 37℃ water bath for 15 min. The digested pancreas was filtered through a 250 μm strainer and washed with GBSS with 0.3% BSA. A gradient was created by resuspending the cells in Nycodenz (VWR 100356–726) and layering in GBSS with 0.3% BSA. Cells were then centrifuged at 1300 x g for 20 min at 4℃. The layer containing PSCs was removed, filtered through a 70 μm strainer, washed in GBSS with 0.3% BSA, and plated for cell culture in DMEM with 10% FBS and penicillin-streptomycin. PSCs were immortalized with TERT and SV40 largeT after several passages.

## Organoid culture

Organoids were isolated from mice bearing PDAC tumors and cultured as previously described (*Boj et al., 2015*). Tumors were minced and digested overnight with collagenase XI (Sigma C9407)

and dispase II (Roche 04942078001) and embedded in 50 µL domes of growth factor reduced (GFR) matrigel (Corning 356231) covered with 500 µL of complete media. Complete media consisted of Advanced DMEM/F12 (Thermo Fisher 12634) containing GlutaMAX (Thermo Fisher 35050), penicillin-streptomycin, HEPES (Thermo Fisher 15630), 0.5 µM TGF-b inhibitor A-83–01 (TOCRIS 2939), 0.05 µg/mL EGF (Thermo Fisher PMG8041), 0.1 µg/mL FGF (Peprotech 100–26), 0.01 µM Gastrin I (TOCRIS 3006), 0.1 µg/mL Noggin (Peprotech 250–38), 10.5 µM Rho Kinase Inhibitor Y-27632 (Sigma Y0503), 1.25 mM N-Acetylcysteine (NAC) (Sigma A9165), 10 mM Nicotinamide (Sigma N0636), 1X B-27 supplement (Thermo Fisher 17504), and 1 µg/mL R-spondin. R-spondin was purified from 293 T cells engineered to produce it using a Protein A Antibody Purification Kit (Sigma PURE1A). Organoids were grown in complete media when passaging. For organoid-PSC co-culture experiments, co-cultures were grown in DMEM without pyruvate (Corning 10–017-CV) supplemented with 10% dialyzed FBS and penicillin-streptomycin. Organoids were regularly tested for mycoplasma contamination using the MycoAlert Plus kit (Lonza) or the Mycoprobe Mycoplasma Detection Kit (R and D Systems).

Organoids were digested to single cells by incubating with 2 mg/mL dispase in Advanced DMEM/F12 with penicillin-streptomycin, HEPES, and GlutaMAX at 37˚C for 20 min. Organoids were then triturated with a fire-polished glass pipette and enzymatically digested with 1 mL TrypLE Express (Thermo Fisher 12605–010) for 10 min rotating at 37˚C, followed by addition of 1 mL of dispase containing media and 10 µL of 10 mg/mL DNAse (Sigma 4527) and digested rotating at 37˚C for 20 min or until single cells were visible under a microscope. Cells were counted and plated in GFR matrigel at a concentration of 2000 cells/well.

## Proliferation assays

50,000 cells were seeded in six-well plates in 2 mL DMEM with 10%FBS and penicillin-streptomycin. The next day, cells were counted for day 0 and media was changed on remaining cells. 8 mL of media was added and cells were left to proliferate for 3 days. On day 3, cells were trypsinized and counted. Alternatively, proliferation was measured using sulforhodamine B staining as previously described (*Vichai and Kirtikara, 2006*). Cells were fixed on day 0 and day 3 with 500 µl of 10% trichloroacetic acid (Sigma T9159) in 1 mL media and incubated at 4˚C for at least 1 hr. Plates were washed under running water and cells were stained with 1 mL sulforhodamine B (Sigma 230162) and incubated at room temperature for 30 min. Dye was removed and cells were washed three times with 1% acetic acid. Plates were then dried and 1 mL of 10 mM Tris pH 10.5 was added to each well to solubilize the dye. 100 µL of each sample was then transferred to a 96-well plate and absorbance was measured at 510 nm on a microplate reader.

## Protein synthesis assays

A fluorescent reporter in which BFP is fused to an unstable *E. coli* dihydrofolate reductase (DHFR) degron domain which is stabilized by trimethoprim (*Han et al., 2014*) was used to determine global protein synthesis rate as previously described (*Darnell et al., 2018*). Briefly, PDAC/PSC cell lines and PDAC organoids expressing the reporter were generated by lentiviral transduction and puromycin selection followed by flow cytometry-based sorting for populations that were BFP-positive after TMP addition for 24–48 hr. For each experiment, the reporter protein was stabilized upon addition of 10 uM trimethoprim (TMP) and fluorescence accumulation was measured in cells or organoids by flow cytometry over several time points within 12 hr of TMP addition. Data were normalized to no TMP controls. Puromycin incorporation assays were performed as previously described (*Schmidt et al., 2009*). 10 µg/mL puromycin was spiked into the medium of cells grown in 6 cm plates. Plates were kept at 37˚C for indicated pulse times (spanning 2.5 to 20 min). As a negative control, 100 µg/mL cycloheximide was added to a plate of cells for 45 min before the addition of puromycin. At the end of the pulse, plates were washed once with ice cold PBS on ice and flash frozen in liquid nitrogen. Cells were harvested from frozen plates by scraping into RIPA buffer containing cOmplete Mini EDTA-free Protease Inhibitor Cocktail (Roche 11836170001) and PhosSTOP Phosphatase Inhibitor Cocktail Tablets (Roche 04906845001) and protein concentration was quantified using the Pierce BCA Protein Assay Kit (Pierce 23225). 2 µL of lysate (approximately 2 µg) was spotted directly onto 0.2 µm nitrocellulose membranes and blotted with primary antibodies against

puromycin (Sigma MABE343 1:25,000 dilution) and vinculin (Abcam ab18058, 1:1000 dilution) as a control.

## Generation of PC and ME1 knockdown and knockout cells

CRISPRi knockdown cell lines of PC and ME1 were generated by transfecting cells expressing modified dCas9-KRAB fusion protein, as previously described (Horlbeck et al., 2016). The target sequences used for PC sgRNAs were (PC1- GCGGCGGCCACGGCTAGAGG, PC2- GTGGAGGCAGGGGCCGTCAG), the sequence for non-targeting control was GCGACTAGCGCCATGAGCGG, and the target sequence of ME1 sgRNA was GCCGCAGTGGCCTCCCGGGT. After transfection, cells were selected under 5 ug/ml puromycin. Rescue of CRISPRi knockdown cell lines of PC was performed by re-expressing the cDNA of the rescued gene under a CMV promoter using a custom lentiviral construct generated on VectorBuilder and selected in 500 ug/ml blasticidin. CRISPR knockout cell lines for PC and ME1 were generated using the LentiCRISPRv2 system, as previously described (Sanjana et al., 2014), with guides against the target sequence 5′ CGGCATGCGGGTCGTGCATA 3′ for PC and 5′ GTTTGGCATTCCGGAAGCCA 3′ for ME1. After transfection, cells were selected under 5 µg/ml of puromycin, single-cell cloned, and knockout validation performed using western blot. For organoids, the same vector systems and guide sequences were used. Organoids were transfected with concentrated virus by spinfection for 45 min at room temperature. For CRISPR knockout organoids, organoid cultures were selected under 5 µg/ml of puromycin, digested to single cells, and then single organoids were picked, expanded, and validated using western blot.

Double knockout cell lines for PC and ME1 were generated using pUSPmNG (U6 sgRNA PGK with mNeonGreen, Li et al., 2019) incorporated into cells via electroporation (Amaxa VPI-1005), and selected by FACS using NeonGreen expression. For organoids, double knockout organoids were generated using the LentiCRISPRv2 system and spinfected for 45 min at room temperature. After transfection, cells were selected under 500 ug/ml of blasticidin, digested to single cells, and then single organoids were picked, expanded, and validated using western blot.

## Flow cytometry

Tumors were dissected, minced, and digested rotating for 30 min at 37°C with 1 mg/mL Collagenase I (Worthington Biochemical LS004194), 3 mg/mL Dispase II (Roche 04942078001), and 0.1 mg/mL DNase I (Sigma D4527) in PBS. Following digestion, cells were incubated with EDTA to 10 mM at room temperature for 5 min. Cells were then filtered through a 70 µm strainer and washed twice with PBS. Single cell suspensions were resuspended in flow cytometry staining buffer (Thermo Fisher 00-4222-57) and first stained with 10 µL of CD16/CD32 monoclonal antibody (Thermo Fisher 14-0161-82) for 15 min to block Fc receptors and then stained using with antibodies to CD45-APC-Cy7 (BD 557659) at 1:100 dilution followed by SYTOX Red Dead Cell Stain (Life Technologies S34859) at 1:1000 dilution to visualize dead cells. All antibodies were incubated for 15–20 min on ice and then washed. Cell sorting was performed with a BD FACS Aria and data was analyzed with FlowJo Software (BD).

## Tumor transplantation

100,000 PDAC cells or 100,000 PDAC cells plus 100,000 PSCs in 100 µL PBS were transplanted subcutaneously into the flanks of C57BL/6J mice (000664). Tumors were measured using calipers and volume was calculated using the formula $V = (\pi/6)(L*W^2)$.

For orthotopic transplants, 100,000 PDAC cells in 50 µL PBS were transplanted into the pancreas of C57BL/6J mice (000664) as previously described (Mayers et al., 2014).

## Immunofluorescence

Tumors were fixed in 4% paraformaldehyde (PFA) rotating overnight at 4°C followed by incubation in 30% sucrose in PBS rotating overnight at 4°C. Tumors were then embedded in optimal cutting temperature (OCT) compound and stored at −80°C until sectioning. Sections were stained with antibodies against α-SMA (Sigma F3777, 1:500 dilution), CK19 (Abcam ab133496 1:100), and tdTomato (Rockland 600-401-379, 1:500 dilution) using DAPI as a nuclear stain.

## Immunohistochemistry

Sections from formalin fixed paraffin embedded mouse tissue or a human pancreatic cancer tissue microarray (Biomax PA961e) were stained with antibodies against PC (Santa Cruz sc-67021, 1:50 dilution) or ME1 (Proteintech 16619–1-AP, 1:200 dilution). The human pancreatic cancer tissue microarray was scored independently by a pathologist (O.H.Y.), and assigned scores of 0–4 for both staining intensity and percent of cells positive for expression of the indicated protein.

## qPCR

RNA was isolated from cells using the RNAqueous-Micro Total RNA Isolation Kit (Life Technologies AM1931) and cDNA was made using the iScript cDNA Synthesis Kit (Bio-Rad Laboratories 1708890). qPCR reactions were performed using SYBR Green Master Mix (Sigma L6544) and primers for pyruvate carboxylase (Forward: 5'- GGG ATG CCC ACC AGT CAC T −3', Reverse: 5'- CAT AGG GCG CAA TCT TTT TGA −3'), malic enzyme 1 (Forward: 5'- TGT GGG AAC AGA AAA TGA GGA GTT −3', Reverse: 5'- TCA TCC AGG AAG GCG TCA TAC T −3'), tdTomato (Forward: 5'- AGC AAG GGC GAG GAG GTC ATC −3' Reverse: 5'- CCT TGG AGC CGT ACA TGA ACT GG −3'), α-sma (Forward: 5'- TCC CTG GAG AAG AGC TAC GAA −3' Reverse: 5'- TAT AGG TGG TTT CGT GGA TGC C −3'), vimentin (Forward: 5'- GTA CCG GAG ACA GGT GCA GT- 3', Reverse: 5'- TTC TCT TCC ATC TCA CGC ATC −3'), Trp53 (Forward: 5'- CTC TCC CCC GCA AAA GAA AAA −3', Reverse: 5'- CGG AAC ATC TCG AAG CGT TTA −3'), CD3ε (Forward: 5'- ATG CGG TGG AAC ACT TTC TGG −3', Reverse: 5'- GCA CGT CAA CTC TAC ACT GGT −3'), F4/80 (Forward: 5'- TGA CTC ACC TTG TGG TCC TAA −3', Reverse: 5'- CTT CCC AGA ATC CAG TCT TTC C −3'), or E-Cadherin (Forward: 5'- GCT CTC ATC ATC GCC ACA G- 3', Reverse: 5'- GAT GGG AGC GTT GTC ATT G- 3') using 18S (Forward: 5'- CGC TTC CTT ACC TGG TTG AT −3', Reverse: GAG CGA CCA AAG GAA CCA TA −3') or 36B4 (Forward: 5'- TCC AGG CTT TGG GCA TCA −3', Reverse: 5'- CTT TAT CAG CTG CAC ATC ACT CAG A −3') as controls. Primer sequences are also included in *Supplementary file 1*.

## Western blot

Cell lines were washed with ice-cold PBS and scraped into RIPA buffer containing cOmplete Mini EDTA-free Protease Inhibitor Cocktail (Roche 11836170001) and PhosSTOP Phosphatase Inhibitor Cocktail Tablets (Roche 04906845001). Lysates were then rotated at 4°C for 20 min and centrifuged for 5 min at max speed in a tabletop centrifuge at 4°C. Organoids were resuspended in ice-cold PBS containing cOmplete Mini EDTA-free Protease Inhibitor Cocktail and PhosSTOP Phosphatase Inhibitor Cocktail Tablets (PBS-PPI). Organoids were then centrifuged at 3000xg for 3 min at 4°C and washed two times in ice-cold PBS-PPI. Cell pellets were resuspended in TNET buffer (1% Triton X-100, 150 mM NaCl, 5 mM EDTA, and 50 mM Tris ph 7.5) containing cOmplete Mini EDTA-free Protease Inhibitor Cocktail and PhosSTOP Phosphatase Inhibitor Cocktail Tablets, incubated on ice for 10 min, and passed through a 26 gauge needle three times. Lysates were centrifuged for 10 min at max speed in a tabletop centrifuge at 4°C. Protein concentration was quantified using the Pierce BCA Protein Assay Kit (Pierce 23225). Western blots were performed using primary antibodies against PC (Santa Cruz sc-271493, 1:100 dilution), ME1 (Proteintech 16619–1-AP, 1:250 dilution), or β-actin (Cell Signaling Technologies 8457, 1:10,000 dilution).

### Quantification and statistical analysis

GraphPad Prism software was used for statistical analysis. All statistical information is described in the figure legends.

## Acknowledgements

We thank the Koch Institute Swanson Biotechnology Center for assistance with flow cytometry, histology, and immunohistochemistry, as well as the members of the Vander Heiden lab for helpful discussions. ANL was a Robert Black Fellow of the Damon Runyon Cancer Research Foundation, DRG-2241–15, and was supported by a NIH Pathway to Independence Award (K99CA234221). ZL and KMS were supported by NIH training grant T32GM007287. AMD acknowledges support from the Jane Coffin Childs Memorial Fund for Medical Research. RF acknowledges support from Swedish

Foundation for Strategic Research, the Knut and Alice Wallenberg Foundation, and the Barbro Osher Pro Suecia Foundation. VG was supported by a Jane Coffin Childs Memorial Fund Postdoctoral Fellowship and NCI TMEN grant U54 CA163109. SS and ECL are supported by the Damon Runyon Cancer Research Foundation (DRG-2367–19, DRG-2299–17). GB was a fellow of the Human Frontiers Science Program (LT000195/2015 L) and EMBO (ALTF 1203–2014). TJ is a Howard Hughes Medical Institute Investigator, David H Koch Professor of Biology, and a Daniel K Ludwig Scholar. NJM is supported by the MRC (CSF MR/P008801/1), NHSBT (WPA15-02) and the NIHR Cambridge BRC. OHY acknowledges support from NIH (R01CA211184, R01CA034992). MGVH acknowledges support from the Lustgarten Foundation, a Faculty Scholar grant from the Howard Hughes Medical Institute, SU2C a division of the Entertainment Industry Foundation, the MIT Center for Precision Cancer Medicine, the Ludwig Center at MIT, the Emerald Foundation, and the NCI (R01CA168653, R01CA201276, R35CA242379, P30CA14051).

## Additional information

### Competing interests

Matthew G Vander Heiden: Reviewing editor, *eLife*. MGVH is a member of the scientific advisory board member for Agios Pharmaceuticals, Aeglea Biotherapeutics, and iTeos Therapeutics, and a co-founder of Auron Therapeutics. Tyler Jacks: TJ is a member of the Board of Directors of Amgen and Thermo Fisher Scientific, is a co-Founder of Dragonfly Therapeutics and T2 Biosystems, and is a scientific advisor of SQZ Biotech, and Skyhawk Therapeutics. The other authors declare that no competing interests exist.

### Funding

| Funder | Grant reference number | Author |
|---|---|---|
| Damon Runyon Cancer Research Foundation | DRG-2241-15 | Allison N Lau |
| Damon Runyon Cancer Research Foundation | DRG-2367-19 | Sharanya Sivanand |
| Damon Runyon Cancer Research Foundation | DRG-2299-17 | Evan C Lien |
| National Cancer Institute | K99CA234221 | Allison N Lau |
| National Institutes of Health | T32GM007287 | Zhaoqi Li<br>Kiera M Sapp |
| Jane Coffin Childs Memorial Fund for Medical Research | | Alicia M Darnell<br>Vasilena Gocheva |
| Swedish Foundation for Strategic Research | | Raphael Ferreira |
| Knut and Alice Wallenberg Foundation | | Raphael Ferreira |
| Barbro Osher Pro Suecia Foundation | | Raphael Ferreira |
| National Cancer Institute | U54CA163109 | Vasilena Gocheva |
| Human Frontier Science Program | LT000195/2015-L | Giulia Biffi |
| EMBO | ALTF 1203-2014 | Giulia Biffi |
| Howard Hughes Medical Institute | | Tyler Jacks<br>Matthew G Vander Heiden |
| MRC | CSF MR/P008801/1 | Nicholas J Matheson |
| NHSBT | WPA15-02 | Nicholas J Matheson |
| NIHR Cambridge BRC | | Nicholas J Matheson |
| National Institutes of Health | R01CA211184 | Omer Yilmaz |

| National Institutes of Health | R01CA034992 | Omer Yilmaz |
| Lustgarten Foundation | | Matthew G Vander Heiden |
| Stand Up To Cancer | | Matthew G Vander Heiden |
| MIT Center for Precision Cancer Medicine | | Matthew G Vander Heiden |
| Ludwig Center at MIT | | Tyler Jacks Matthew G Vander Heiden |
| Emerald Foundation | | Matthew G Vander Heiden |
| National Cancer Institute | R01CA168653 | Matthew G Vander Heiden |
| National Cancer Institute | R01CA201276 | Matthew G Vander Heiden |
| National Cancer Institute | R35CA242379 | Matthew G Vander Heiden |
| National Cancer Institute | P30CA14051 | Matthew G Vander Heiden |

The funders had no role in study design, data collection and interpretation, or the decision to submit the work for publication.

## Author contributions

Allison N Lau, Conceptualization, Funding acquisition, Investigation, Visualization, Methodology, Writing - original draft, Writing - review and editing; Zhaoqi Li, Conceptualization, Investigation, Writing - review and editing; Laura V Danai, Anna M Westermark, Alicia M Darnell, Raphael Ferreira, Vasilena Gocheva, Sharanya Sivanand, Evan C Lien, Kiera M Sapp, Christopher R Chin, Shawn M Davidson, Omer Yilmaz, Investigation, Writing - review and editing; Jared R Mayers, Giulia Biffi, Nicholas J Matheson, Methodology, Writing - review and editing; David A Tuveson, Supervision; Tyler Jacks, Supervision, Writing - review and editing; Matthew G Vander Heiden, Conceptualization, Supervision, Funding acquisition, Methodology, Writing - original draft, Writing - review and editing

## Author ORCIDs

Allison N Lau https://orcid.org/0000-0003-4250-7355
Raphael Ferreira http://orcid.org/0000-0001-9881-6232
Jared R Mayers http://orcid.org/0000-0002-8607-1787
Nicholas J Matheson http://orcid.org/0000-0002-3318-1851
Matthew G Vander Heiden https://orcid.org/0000-0002-6702-4192

## Ethics

Animal experimentation: All animal studies were approved by the MIT Committee on Animal Care under protocol #0119-001-22.

## Decision letter and Author response

Decision letter https://doi.org/10.7554/eLife.56782.sa1
Author response https://doi.org/10.7554/eLife.56782.sa2

# Additional files

## Supplementary files

- Supplementary file 1. qPCR primer sequences. Sequences of primers used for qPCR reactions.

- Transparent reporting form

## Data availability

All data generated or analyzed during this study are included in the manuscript and supporting files.

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
