## [Decision Letter]

**Acceptance summary:**

The manuscript describes a technique that capitalizes on the relatively slow turnover of macromolecules to infer metabolic differences between different cell types within complex tissues, including tumor organoids and intact tumors. The authors use this technique to demonstrate the enhanced use of pyruvate carboxylation in cancer cells relative to fibroblasts, and to show that apparent pyruvate carboxylation uses magic enzyme as well as the more conventional anaplerotic enzyme pyruvate carboxylase.

**Decision letter after peer review:**

Thank you for submitting your article "Dissecting cell type-specific metabolism in pancreatic ductal adenocarcinoma" for consideration by *eLife*. Your article has been reviewed by three peer reviewers, including Ralph DeBerardinis as the Reviewing Editor and Reviewer #1, and the evaluation has been overseen by Richard White as the Senior Editor. The following individual involved in review of your submission has agreed to reveal their identity: Joshua D Rabinowitz (Reviewer #3).

The reviewers have discussed the reviews with one another and the Reviewing Editor has drafted this decision to help you prepare a revised submission.

Summary:

The authors describe an isotope infusion method to infer distinct metabolic activities from distinct cell types within complex organoids and tumors. Because macromolecules like proteins turn over much more slowly than intermediary metabolites, the authors reasoned that proteins would better retain labeling features during ex vivo manipulation to separate different cell types. They find that cancer cells display elevated enrichment of glucose-derived nonessential amino acids in proteins compared to stromal and other non-malignant cells. Labeling in aspartate suggested elevated pyruvate carboxylation in cancer cells, and the authors provide evidence that this carboxylation involves two enzymes: pyruvate carboxylase and malic enzyme-1. Knocking out either enzyme reduced tumor growth. Overall, the paper presents an interesting technique and applies it to advance our understanding of pyruvate carboxylation in pancreatic cancer.

Essential revisions:

1) Regarding the rationale for assessing label in amino acids for proteins, the authors should place a stronger emphasis on the "good news" that TCA labeling patterns are reasonably robust to cell handling. The authors might also make more overt whether or not they tested robustness of these labeling patterns to actual sorting, or just cells sitting in a dish. If they did not test sorting, they might emphasize that, at a minimum, the labeling patterns seem robust enough that the field can be relatively confident of the conclusions of prior work with direct metabolism quenching and TCA metabolite measurement (i.e. the terminal events and delays in quenching in these prior studies are unlikely to have impacted TCA labeling patterns).

2) Several questions were raised about the relative importance of ME1 and PC for anaplerosis. First, the reader is left with the impression that PC and ME1 are equivalent in their ability to carboxylate pyruvate for anaplerosis/anabolism. But PC is generally considered a unidirectional reaction whereas ME1 is reversible, and the labeling data are certainly consistent with reversibility. The authors should avoid suggesting that malic enzyme has functional redundancy with pyruvate carboxylase until the net direction of ME1 flux is clarified, or if orthogonal evidence is available to support this argument.

3) Along these lines, the paper does not appear to contain any studies in which ME1 was knocked out but PC left intact. If such data are available (ideally isotope labeling and metabolite levels, but either one would be helpful), this would strengthen the paper. If not, then the authors should be more conservative in their interpretation of ME1's role in anaplerosis.

4) A major long-term question is why pyruvate carboxylase and ME1 are selectively essential in vivo. Why do they become so important when NADPH can also be produced from the oxidative pentose phosphate pathway, and several pathways are available for anaplerosis? This issue deserves more attention in the Discussion.

5) In Figure 1, the authors show labeling of glucose in the plasma, and comment that products of glycolysis (pyruvate, lactate, alanine) are labeled in the tumor. Because these metabolites can be readily taken up by some tissues, their labeling in the blood should also be reported. Labeling in circulating aspartate should also be reported.

6) If available already from some of the MS analyses conducted for labeling, it would be helpful to see more data on metabolite concentrations (relative concentrations would be sufficient), especially for key players such as malate, aspartate, pyruvate and, if appropriate methods were used to acquire the data, NADPH, NADP, NADH and NAD.

7) Glucose enrichment in plasma is provided, but not glucose enrichment in the tumors. If these values differ among the tumor models (WT, KP-/-C, KPC), this might explain some differences in labeling of downstream, metabolites. If these measurements are available, they should be included. Plasma insulin levels in the different models should also be provided if available.

---

## [Author Response]

Essential revisions:1) Regarding the rationale for assessing label in amino acids for proteins, the authors should place a stronger emphasis on the "good news" that TCA labeling patterns are reasonably robust to cell handling. The authors might also make more overt whether or not they tested robustness of these labeling patterns to actual sorting, or just cells sitting in a dish. If they did not test sorting, they might emphasize that, at a minimum, the labeling patterns seem robust enough that the field can be relatively confident of the conclusions of prior work with direct metabolism quenching and TCA metabolite measurement (i.e. the terminal events and delays in quenching in these prior studies are unlikely to have impacted TCA labeling patterns).

This is a reasonable point, and our intention was neither to question prior work examining TCA cycle labeling in sorted cells nor to imply that approach does not generate interesting data. We have modified our Discussion to make this more clear. Nevertheless, we are hesitant to overemphasize that TCA cycle labeling is robust with respect to sorting in all cases. While we agree that in the cells and conditions we tested, some metabolite labeling patterns and levels were robust to cell handling, this was not the case for all TCA metabolites. As reviewed in Buescher et al., 2015, interpretation of isotope labeling often assumes both isotopic and metabolic steady-state where neither levels nor labeling patterns are changing. We are the first to admit that from a practical standpoint, one can still derive interesting information from labeling studies even in conditions where isotopic and metabolic steady-state are not achieved (including in this study where we do not reach steady-state labeling of amino acids in protein). We discuss these points in the revised manuscript and stress that we are not calling into question the conclusions of prior work.

Regarding the question about whether our control experiments considered actual cell sorting or cells siting in a dish, we did not explicitly test how sorting cells affects metabolite labeling or levels. Rather, we focused our control experiments on the effects of processing time as this was most relevant to the data we included in the study. We thank the reviewers for asking us to clarify this important point. In the revised manuscript we discuss that we did not directly test the effect of sorting, but rather the effect of time before cell metabolites are extracted and that any further effects of sorting on free metabolites was not addressed by our analysis.

2) Several questions were raised about the relative importance of ME1 and PC for anaplerosis. First, the reader is left with the impression that PC and ME1 are equivalent in their ability to carboxylate pyruvate for anaplerosis/anabolism. But PC is generally considered a unidirectional reaction whereas ME1 is reversible, and the labeling data are certainly consistent with reversibility. The authors should avoid suggesting that malic enzyme has functional redundancy with pyruvate carboxylase until the net direction of ME1 flux is clarified, or if orthogonal evidence is available to support this argument.

We thank the reviewers for pointing this out, as we did not intend to suggest ME1 and PC are functionally redundant. Please see our response to related point 3 below for a discussion of further data and how we modified the text to correct this issue.

3) Along these lines, the paper does not appear to contain any studies in which ME1 was knocked out but PC left intact. If such data are available (ideally isotope labeling and metabolite levels, but either one would be helpful), this would strengthen the paper. If not, then the authors should be more conservative in their interpretation of ME1's role in anaplerosis.

Again this is an important point and we agree fully with the reviewers’ comments. As the reviewers appreciate, it is difficult to quantitatively determine the flux in cells where both PC and ME1 can contribute to anaplerosis, and agree that these activities are unlikely to be equivalent. Indeed we did perform experiments where ME1 was knocked out and PC left intact. The isotope labeling data from those ME1 null cells unexpectedly showed a larger decrease in labeling consistent with pyruvate carboxylation than was observed in PC null cells (with ME1 left intact). We originally opted not to include these data in the manuscript so as not to mislead readers who might conclude ME1 is more important for anaplerosis than PC. This implication would be quite controversial, and in our view is not a conclusion that should be drawn based only on these data. Furthermore, as discussed in the manuscript, carbon cycling involving both enzymes is likely when both enzymes are present, and dissecting pyruvate cycling is a formidable challenge. This led us to conclude that both PC and ME1 can contribute to anaplerotic TCA flux, but this study cannot distinguish between the relative activities of the two enzymes.

Nevertheless, the point raised by the reviewers is appreciated as we agree this is an obvious experiment some readers might expect, and despite our reservations about interpretation, we concur that isotope-labeling data from ME1 single knockout cells does strengthen the conclusion that ME1 can contribute to TCA cycle anaplerosis in pancreatic cancer. Thus, we have included the isotope-labeling experiments where ME1 was knocked down or knocked out and PC was left intact in Figure 7—figure supplement 1 of the revised manuscript. This includes data showing cells with reduced or absent levels of ME1 have reduced M+1 aspartate labeling from 1-^13^C-pyruvate. We also include a discussion that these data should not necessarily be used to conclude that flux through ME1 is higher than PC, particularly in cells where both enzymes are expressed.

Functional experiments in which ME1 was knocked out with PC left intact remain in Figure 7 of the manuscript. The effects of ME1 knockout show a similar effect to PC knockout, with minimal effects on cell proliferation in culture (Figure 7F), but a reduction in organoid growth (Figure 7G-H) and tumor formation (Figure 7I).

4) A major long-term question is why pyruvate carboxylase and ME1 are selectively essential in vivo. Why do they become so important when NADPH can also be produced from the oxidative pentose phosphate pathway, and several pathways are available for anaplerosis? This issue deserves more attention in the Discussion.

We agree that this is an interesting topic and will be an important question for future research. We now include additional discussion of the selective in vivo essentiality of PC and ME1 in the Discussion section of the revised manuscript.

5) In Figure 1, the authors show labeling of glucose in the plasma, and comment that products of glycolysis (pyruvate, lactate, alanine) are labeled in the tumor. Because these metabolites can be readily taken up by some tissues, their labeling in the blood should also be reported. Labeling in circulating aspartate should also be reported.

We agree that labeling of plasma metabolites other than glucose is informative and should be reported. We apologize for this oversight in the original manuscript and now include these data in Figure 1—figure supplements 1-3 of the revised manuscript and in the source data files. Regarding the labeling of circulating aspartate, we point out that prior work using this model has shown that aspartate is not accessible to these cancer cells (Sullivan et al., 2018) and so plasma labeling from this metabolite will minimally contribute to labeling in PDAC tumors.

6) If available already from some of the MS analyses conducted for labeling, it would be helpful to see more data on metabolite concentrations (relative concentrations would be sufficient), especially for key players such as malate, aspartate, pyruvate and, if appropriate methods were used to acquire the data, NADPH, NADP, NADH and NAD.

All mass spectrometry data were collected using GC-MS and therefore unfortunately NADPH, NADP, NADH, and NAD were not measured in this study. We also agree that relative concentrations of metabolites are important to include and we apologize for omitting these data from the original manuscript. Relative levels of metabolites measured from the infusion experiments are now included in Figure 1—figure supplements 1-2 and in the source data files for the revised manuscript.

7) Glucose enrichment in plasma is provided, but not glucose enrichment in the tumors. If these values differ among the tumor models (WT, KP-/-C, KPC), this might explain some differences in labeling of downstream, metabolites. If these measurements are available, they should be included. Plasma insulin levels in the different models should also be provided if available.

Unfortunately we did not measure glucose enrichment in tumors for these infusion experiments. Since glucose is thought to be rapidly metabolized after entry into tumor cells, we expect that any measurements of labeled glucose from tumor tissue to reflect labeling of glucose in plasma as has been observed in other contexts, but point out in the revised manuscript that this was not explicitly confirmed in this study.

We also did not measure plasma insulin as part of these experiments; however, in our previous work plasma insulin levels were measured in PDAC tumor bearing mice from the same KP^-/-^C mouse model and compared to levels in wild-type mice. In those studies, no significant differences in plasma insulin levels were observed in either the fed or fasted states, or after a glucose injection which substantially raises glucose levels, between the control and tumor bearing animals (Danai et al., 2018).